



# 1 To quantify the impact of SST feedback periodicity on

# 2 atmospheric intraseasonal variability in the tropical regions

Yung-Yao Lan[1], Huang-Hsiung Hsu[1] and Wan-Ling Tseng[2]
[1]Research Center for Environmental Changes, Academia Sinica, Taipei 11529, Taiwan
[2]International Degree Program in Climate Change and Sustainable Development, National Taiwan
University,Taipei 10617, Taiwan
*Correspondence to*: Huang-Hsiung Hsu (hhhsu@gate.sinica.edu.tw)





**Abstract**

9        This study couples a high-resolution 1-D TKE ocean model (the SIT model) with

the Community Atmosphere Model 5.3 (CAM5.3; hereafter CAM5–SIT)
configuration, to highlight significant experiments that investigate the influence of
different periods of sea surface temperature (SST) feedback, such as 30 minutes, 1, 3,
6, 12, 18, and 30 days, on the Madden-Julian Oscillation (MJO). It substantially
breaks through the limitations of flux coupler through air–sea coupling. The aim is to
assess the scientific reproducibility and consistency of the findings across different
SST feedback cycles in the field of modeling science. Comparing the results to the
fifth generation ECMWF reanalysis (ERA5), the high-frequency experiments (C–
CTL, C–1day, and C–3days) and low-frequency experiments (C–6days, C–12days,
and C–18days) exhibit higher fidelity in capturing various aspects of MJO, except for
the C–30days experiment. These aspects in characterizing the basic features of the
MJO such as encompass intraseasonal periodicity, eastward propagation, coherence in
the MJO band, tilting vertical structure, the lead–lag relationship between MJO-
related atmosphere and SST variation, phase 2 column-integrated moisture static
energy (MSE) tendency, and the projection of all MSE terms onto the MSE tendency
of ERA5 across the Maritime Continent (MC). The MJO simulation performance of
this study can be assessed in two ways. Firstly, the high-frequency experiments
generally capture MJO characteristics, albeit with slightly lower results compared to
ERA5 and NOAA data. Secondly, the low-frequency experiments show robust MJO
simulations, which can be attributed to the accumulation of energy (temperature) in
the upper ocean. This leads to the accumulation of shortwave and longwave
radiations, as well as surface heat fluxes from the atmosphere.



## 1. Introduction

The Madden–Julian Oscillation (MJO) is a large-scale tropical circulation that propagates eastward from the tropical Indian Ocean (IO) to the western Pacific (WP) with a periodicity of 30–80 days (Madden and Julian, 1972). In the Indo-Pacific region, there is increasing evidence that MJO processes are involved in intraseasonal variability of sea surface temperature (SST) (Chang et al., 2019; DeMott et al., 2014, 2015; Jiang et al., 2015, 2020; Krishnamurti et al., 1998; Li et al., 2014; Li et al., 2020a; Newman et al., 2009; Pei et al., 2018; Stan, 2018; Tseng et al., 2015). These studies confirm that including air–sea interactions significantly improves the simulation of the MJO.

The ocean's response to intraseasonal atmospheric variability, such as surface shortwave radiation, turbulent heat fluxes controlled by wind speed, and ocean processes driven by wind stress, plays a crucial role in causing intraseasonal SST variability related to the MJO (Li et al., 2014). Incorporating two-way coupling between the ocean and atmosphere is expected to be valuable for simulating and predicting intraseasonal variability (e.g., DeMott et al., 2014; Lan et al., 2022; Stan, 2018; Tseng et al., 2015, 2020). However, the influence of sub-seasonal (e.g., beyond a phase) air–sea coupling on convection and related oceanic features is still not fully understood.

In this study, we aim to investigate the specific effects of oceanic feedback frequency (FF) in sub-seasonal periodicity through air–sea coupling on the eastward propagation of the MJO as simulated by the Community Atmosphere Model 5.3 (CAM5.3) coupled with the one-column ocean model Snow–Ice–Thermocline (SIT), referred to as CAM5–SIT (Lan et al., 2022). The tropical air–sea interaction, influenced by the upper ocean, plays a crucial role in determining MJO characteristics





due to the high heat capacity of the upper ocean, which acts as a significant source of
heat energy for atmospheric variability (Liang and Du, 2022). The SIT model,
consisting of 41 vertical layers, enables the simulation of SST and upper-ocean
temperature variations with high vertical resolution. The vertical resolution is set to
12 layers within the first 10.5 m and 6 layers between 10.5 m and 107.8 m. The fine
resolution simulates the upper ocean warm layer, which includes a layer at 0.05 mm
that replicates the cool skin of the ocean surface (Tseng et al., 2015, 2020; Lan et al.,
2010, 2022). Previous studies have emphasized the importance of vertical resolution
in accurately simulating the MJO, with Tseng et al. (2015) demonstrating the
necessity of a 1 m vertical resolution when coupling SIT with the European
Centre/Hamburg Model version 5 (ECHAM5), referred to as ECHAM5–SIT in the
tropics. Furthermore, Shinoda et al. (2021) indicated the positive impact of high
vertical resolution near the ocean surface on MJO prediction abilities based on
simulations of the National Oceanic and Atmosphere Administration (NOAA)
Climate Forecast System. Ge et al. (2017) highlighted the presence of a high vertical
temperature gradient within the upper 10 m of the MJO event, particularly during dry
and clear periods, underscoring the need for a resolution of approximately 1 m to
accurately capture intraseasonal SST variability.

Several recent studies have made significant progress in understanding the

impact of air–sea coupling on the MJO, particularly at sub-daily scales (e.g., DeMott
et al., 2015; Kim et al., 2018; Seo et al., 2014; Voldoire et al., 2022; Zhao and
Nasuno, 2020). However, there is relatively limited discussion on air–sea coupling at
the sub-seasonal scale. Several studies have undertaken investigations into the impact
of intraseasonal SST on the MJO by conducting various model experiments,
encompassing both coupled and uncoupled models. (e.g., DeMott et al., 2014; Gao et
al., 2020b; Klingaman, and Demott, 2020; Pariyar et al., 2023; Stan, 2018). Stan





(2018) found that in the air–sea coupling run, the peak in surface fluxes (latent heat
and sensible heat) is stronger and occurs earlier compared to the uncoupling run with
sub-5-day SST variability. Additionally, the absence of 1–5-day variability in SST
promotes the amplification of westward power associated with Rossby waves. Based
on these modeling studies, it is concluded that the atmospheric response to sub-
seasonal SST variances can be determined. Replay simulations using time-varying
coupled global climate model (CGCM) SSTs as atmospheric general circulation
model (AGCM) boundary conditions showed a reduced dynamic range of SST
anomalies, leading to weakened air–sea heat fluxes and eastward propagation
(DeMott et al., 2014; Gao et al., 2020b; Klingaman, and Demott, 2020; Pariyar et al.,
2023). Stan (2018) also demonstrated that eliminating 1–5-day variability of surface
boundary forcing reduces the intraseasonal variability of the tropics during boreal
winter in the case of CGCM SSTs as AGCM boundary conditions. However, the
effect of sub-seasonal SST variances on the MJO is still not fully understood in both
coupled and uncoupled experiments.
As demonstrated by Fu et al. (2017), underestimation (overestimation) of the
air–sea coupling's impact on MJO simulations occurs when there is weakness (strong)
in the intraseasonal SST anomaly. Understanding the manifestation of heat fluxes in
the significant intraseasonal oscillation (ISO) is crucial for the development of
intraseasonal SST variability (Liang et al., 2018). In aquaplanet simulations
conducted by Arnold et al. (2013), where equatorial SST were set at approximately
26°, 29°, 32°, and 35° C, it was observed that the intraseasonal variance
(wavenumbers 1–3, periods 20–100 days) exhibited a significant and consistent
increase with increasing SST. Furthermore, Savarin and Chen (2022) have found that
improved air–sea heat fluxes result in systematic enhancements in precipitation,
winds, SST, and the mixed layer in the ocean. Hence, the combination of higher



tropical SST and enhanced SST variances collectively contribute to an intensified
eastward propagation of the MJO.

Improvements in the MJO in coupled simulations can be attributed to several

factors. Firstly, enhanced low-level convergence and convective instability to the east
of convection, as well as enhanced latent heat fluxes and SST cooling to the west of
convection, contribute to the improved eastward propagation and regulation of MJO
periodicity (DeMott et al., 2014). SST gradients have been found to induce patterns of
mass convergence and divergence within the marine boundary layer (MBL), initiating
atmospheric convection (de Szoeke and Maloney, 2020; Lambaerts et al., 2020). The
basic state SST or basic state moist static energy (MSE) plays a crucial role in MJO
instability (Wang et al., 2016). Moisture convergence in the MBL accumulates MSE
and increases convective instability to the east of the main convection, facilitating the
eastward propagation of the MJO (Hsu and Li, 2012; Wang et al., 2016). The
increased low-level convergence is associated with shallow convection induced by
SST-induced convective instability (DeMott et al., 2014). An analysis of MSE
convergence by de Szoeke and Maloney (2020) demonstrates that intraseasonal SST
fluctuations drive the overall MSE tendency, contributing to MJO generation and
propagation. Arnold et al. (2013) demonstrate that the vertical advection projection
exhibits a positive trend. Specifically, they find that it acts as a strong damping
mechanism at low SST, whereas it transitions into an energy source at high SST.
Based on this observation, we can infer that alterations in vertical MSE advection are
probably accountable for the amplified variability of the MJO in relation to SST.

The structure of this paper is organized as follows. Section 2 introduces the

models, data, methodologies, and experiments employed in this study. The
performance of the CAM5–SIT models in simulating the MJO is discussed in Section
3, while Section 4 focuses on the impact of different sub-seasonal periodicity



configurations on MJO simulations through detailed MJO diagnostics. Finally,
Section 5 presents the conclusions drawn from this study.

**2. Data, model experiments, and methodology**
**2.1 Observational, atmospheric, and oceanic data**

An observational data set used in this study includes precipitation from the

Global Precipitation Climatology Project (GPCP, 1° resolution, 1997–2010; Adler et
al., 2003), outgoing longwave radiation (OLR, 1° resolution, 1997–2010; Liebmann,
1996), and daily SST (optimum interpolated SST, OISST, 0.25° resolution, 1989–
2010; Banzon et al., 2014) from the National Oceanic and Atmosphere
Administration.

The atmospheric variables used in this study were obtained from the fifth-

generation reanalysis of the European Centre for Medium-Range Weather Forecasts
(ECMWF) known as the fifth generation ECMWF reanalysis (ERA5), with a
resolution of 0.25° for the period of 1989–2020 (Hersbach and Dee, 2016). Various
variables from ERA5 were considered, including zonal wind, meridional wind,
vertical velocity, temperature, specific humidity, sea level pressure, geopotential
height, latent heat, sensible heat, and shortwave and longwave radiation. For the
initial conditions of a SIT model, SST data was obtained from the Hadley Centre Sea
Ice and Sea Surface Temperature dataset version 1 (HadISST1), with a resolution of
1° for the period of 1982–2001 (Rayner et al., 2003). The ocean subsurface data,
including climatological ocean temperature, salinity, and currents in 40 layers, were
retrieved from the National Centers for Environmental Prediction (NCEP) Global
Ocean Data Assimilation System (GODAS) with a resolution of 0.5° for the period of
1980–2012 (Behringer and Xue, 2004). These data were used for nudging in the SIT
model.




**2.2 Experimental design**

In this study, we investigate the role of oceanic FF in sub-seasonal periodicity

using the coupled model CAM5–SIT and the uncoupled AGCM (A–CTL). Previous
studies (Lan et al., 2022; Tseng et al., 2022) have provided a detailed description of
the every timestep coupling CAM5–SIT model and its performance in simulating
atmospheric variability and the MJO. This study involved a series of 30-year
numerical experiments (as shown in Table 1). We overcame the limitations of the
National Center for Atmospheric Research (NCAR) Climate System Model (CSM)
Flux Coupler (CPL) by implementing similarly asymmetric exchange frequencies
between the atmosphere and the ocean. The SST value is fixed at each timestep within
the experimental periodicity through a straightforward approach to create various
intraseasonal SST (e.g., 30 minutes, 1, 3, 6, 12, 18, and 30 days) feedback
atmospheric experiments. It is important to note that every timestep involves
bidirectional interaction in the CPL.

Two sets of experiments were conducted, each representing a different SST

feedback frequency:

(1) The high-frequency SST feedback set: This set includes the control

experiment (C–CTL) with SST feedback at every timestep (FF as 48/day), as

well as experiments with SST feedback once a day (C–1day: FF as 1/day)

and every 3 days (C–3days: FF as 1/3days), respectively.

(2) The low-frequency SST feedback set: This set includes experiments with

SST feedback returning to the atmosphere every 6 days (C–6days: FF as

1/6days), 12 days (C–12days: FF as 1/12days), 18 days (C–18days: FF as

1/18days), and 30 days (C–30days: FF as 1/30days), respectively.

In all experiments, there is a common configuration: CAM5 forces SIT at each



timestep, SIT has the same vertical resolution, and coupling is implemented between
30° N to 30° S in the entire tropics. The only difference lies in the frequency of SIT's
SST feedback into the atmosphere. This choice is driven by two factors related to
tropical coupling.

Firstly, the MJO predominantly occurs in tropical regions (Jiang et al., 2020;

Kang et al., 2020; Shinoda et al., 2021), hence coupling was specifically implemented
between 30° N to 30° S. This focuses the coupling on the region where the MJO is
most active.

Secondly, coupling a one-dimensional ocean model without surface flux

correction to the extratropics would neglect the influence of strong ocean currents,
such as the Kuroshio and Gulf Streams, leading to significant biases. Therefore,
coupling is limited to the tropical region to avoid these biases and ensure a more
realistic representation of the air–sea interactions relevant to the MJO.

Forcing of the coupled and uncoupled models' initial conditions was done using

a climatological monthly-mean HadSST1 dataset. The monthly Global Ocean Data
Assimilation System (GODAS) dataset was linearly interpolated to obtain daily
values of oceanic temperature, salinity, u-current, and v-current for nudging purposes.
The ocean was weakly nudged (using a 30-day time scale) between depths of 10.5 m
and 107.8 m, and strongly nudged (using a 1-day time scale) below 107.8 m, based on
the climatological ocean temperature data from NCEP GODAS. No nudging was
applied in the upper-most 10.5 meters.

During the simulation, the SIT recalculated the SST within the tropical air–sea

coupling region, which spans from 30° S to 30° N. Outside this coupling region, the
prescribed annual cycle of HadSST1 was used. The ocean bathymetry for the SIT was
derived from the NOAA ETOPO1 data (Amante and Eakins, 2009). To ensure
consistency and comparability, all observational, atmospheric, oceanic, and reanalysis





data were interpolated into a horizontal resolution of 1.9° × 2.5° for model
initialization, nudging, and comparison of experimental simulations.

**2.3 Methodology**
The analysis focused primarily on the boreal winter period (November–April),
which exhibits the most pronounced eastward propagation of the MJO. To identify
intraseasonal variability, the CLIVAR MJO Working Group diagnostics package
(CLIVAR, 2009) and a 20–100-day filter (Wang et al., 2014) were employed. MJO
phases were defined following the index (namely, RMM1 and RMM2) proposed by
Wheeler and Hendon (2004), which utilizes the first two principal components of
combined near-equatorial OLR and zonal winds at 850 and 200 hPa. The band-pass
filtered data were employed to calculate the index and define the MJO phases.
Furthermore, an analysis of column-integrated MSE budgets was conducted to
investigate the association between the tropical convection and large-scale
circulations. The column-integrated MSE budget equation (e.g., Sobel et al., 2014) is
approximately given by
$$\langle\frac{\partial h}{\partial t}\rangle' = -\langle u\frac{\partial h}{\partial x}\rangle' - \langle v\frac{\partial h}{\partial y}\rangle' - \langle w\frac{\partial h}{\partial p}\rangle' + \langle LW\rangle' + \langle SW\rangle' + \langle SH\rangle' + \langle LH\rangle' \qquad (1)$$
where $h$ denotes the moist static energy;
$$h = c_p T + gz + L_v q \qquad (2)$$
where $T$ is temperature (K); $q$ is specific humidity (Kg Kg$^{-1}$); $c_p$ is dry air heat
capacity at constant pressure (1004 J K$^{-1}$ kg$^{-1}$); $L_v$ is latent heat of condensation
(taken constant at $2.5 \times 10^6$ J kg$^{-1}$); $u$ and v are horizontal and meridional velocities
(m s$^{-1}$), respectively; $\omega$ is the vertical pressure velocity (Pa s$^{-1}$); $LW$ and $SW$ are the
longwave and shortwave radiation fluxes (W m$^{-2}$), respectively; and $LH$ and $SH$ are
the latent and sensible surface heat fluxes (W m$^{-2}$), respectively. The angle brackets





(⟨∗⟩) represent mass-weighted vertical integration from 1000 to 100 hPa; and the
intraseasonal anomalies are represented as ⟨∗⟩′, which were isolated using a 20–100-
day bandpass filter (Wang et al., 2014).

**3.    Results**
**3.1 The basic state SST variability**

The variability of SSTs plays a crucial role in the dynamics of the MJO. In the

absence of interseasonal SST variability, as observed in the uncoupled A–CTL
simulations, the eastward propagation of the MJO is disrupted, resulting in weakened
or fragmented MJO activity. Studies based on observations from TOGA COARE and
DYNAMO have revealed that MJO events exhibit a stronger ocean temperature
response compared to average conditions (de Szoeke et al., 2014). Interseasonal SST
variability in the tropics, resulting from air–sea coupling, significantly impacts the
behavior of the MJO behavior and atmospheric circulation. Warmer SSTs to the east of
convection enhance the release of latent heat, triggering atmospheric convection and
strengthening the MJO. Conversely, cooler SSTs in this region create a more stable
atmospheric environment, which is less favorable for the development and propagation
of the MJO (DeMott et al., 2015). The activity and strength of the MJO are influenced
by SST in the region. Cooler than average sea surface temperatures (SSTs) in this
region are associated with the passage of MJO activity and a tendency towards
decreased intensity.

Table 2 presents the oceanic temperature anomalies for the DJF seasonal mean,

including the differences in oceanic temperature between the SST and depths of 10m
($\overline{\Delta T}_{0-10m}$) and 30m ($\overline{\Delta T}_{0-30m}$), as well as phase anomalies of 20–100 days maximum
and minimum SST and oceanic temperature at 10m depth ($T_{10m}$). The region of 110–
130° E and 5–15° S was selected because it shows the largest variation in the 20–100-
day bandpass-filtered SST when the MJO passes over the Indo-Pacific region. Except





for C–30days, the DJF seasonal mean SST shows a slight increase with the higher SST
feedback periodicity, while the SST standard deviation remains within 0.8 K. In the
critical region (110–130° E, 5–15° S), experiments with high frequency SST feedback
periodicity exhibit a mean SST of less than 1.4 K during DJF, while experiments with
low frequency SST feedback periodicity range from -1.0 K to 0.5 K compared to the
OISST dataset.
Understanding the variations in SST during DJF in the Indo-Pacific region is
critical for predicting and interpreting the MJO's behavior. The temperature differences
between observed monthly mean SST and NCEP GODAS reanalysis data ($\overline{\Delta T}_{0-10m}$
and $\overline{\Delta T}_{0-30m}$) as well as AGCM are not compared here. The $\overline{\Delta T}_{0-10m}$ in high-
frequency experiments maintain 0.1K temperature difference. In low-frequency
experiments, $\overline{\Delta T}_{0-10m}$ increase from 0.1 to 1.0 K as SST feedback periodicity
increases correspondingly. The temperature difference ($\overline{\Delta T}_{0-30m}$) in both high-
frequency and low-frequency experiments remains approximately 0.8K, except for C–
30days. In the daily OISST SST phase anomalies, the maximum and minimum values
are approximately maintained at $\pm 0.2$K. However, compared to OISST or model
simulations, the uncoupled A–CTL, which uses monthly mean OISST, shows
significantly smaller mean anomalies in the SST phase, on the order of 1–2 magnitudes
smaller. In the high-frequency experiments, SST phase anomalies exhibit similar
magnitudes of $\pm 0.2$K as observed. The SST means in both the high-frequency and
low-frequency experiments reach their maximum in phase 3, lagging about 1 phase
behind the OISST. The maximum and minimum $T_{10m}$ values indicate that the
atmospheric heat/cooling ocean process is consistently mixed in the C–CTL
experiment, but not in the low-frequency experiments.
According to CLIVAR diagnostics, there are diverse behaviors observed in MJO
simulations, as indicated by the slight difference between phase anomalies of C–3days





maximum SST and $T_{10m}$ compared to C–CTL and C–1day, which indicates diverse
behaviors of MJO simulations, according to CLIVAR diagnostics. Fu et al. (2017)
indicated that too weak intraseasonal SST anomaly in coupled models would lead to
the underestimation of the impacts of air–sea coupling on MJO simulations.

**3.2 MJO simulation: high-frequency and low-frequency SST feedback**
**experiments**
**3.2.1   General structure**
We conducted SST feedback experiments with high-frequency and low-frequency
responses, as well as uncoupled AGCMs, to compare the simulated characteristics of
the MJO. The propagation characteristics of the different experiments were analyzed
using the wavenumber-frequency spectrum (W-FS). The spectra of unfiltered U850 in
ERA5 reanalysis, A–CTL, C–CTL, C–1day, C–3days, C–6days, C–12days, C–
18days, and C–30days are shown in Fig. 1a–i, respectively. The C–CTL experiment
accurately captures the eastward propagating signals at zone wavenumber 1 and for
periods of 30 to 80 days (Fig. 1a and 1c), although with a slightly larger amplitude
than ERA5. However, the uncoupled A–CTL produces an unrealistic spectral shift to
time scales longer than 30–80 days (Fig. 1b) and exhibits westward propagation at
wavenumber 2.
C–3days tend to reduce the interseasonal variability of the MJO compared to the
C–CTL experiment under coupled runs, which is consistent with the results of Stan
(2018) in uncoupled experiments that force the atmosphere by surface boundary. de
Boisséson et al. (2012) also found that hindcasts of the MJO are sensitive to changes
in SST boundary conditions, although daily and weekly SST forecasts are similar. In
this study, it was observed that the high-frequency experiments limited the variance of
the MJO. According to Stan (2018), the lack of 1–5-day SST variability favors an
increase in westward power associated with Rossby waves. The W-FS of the C–1day





experiment showed two peaks for zone wavenumber 1 over the 30 to 80-day period.
This might be attributed to the inconsistency in day and night variations when the SST
feedback of C–1day is returned to the atmosphere at different locations. Except for C–
30days, the low-frequency experiments enhance the W-FS of U850 during
interseasonal periods. In this study, low-frequency SST variability is not enhanced in
the unrealistic westward W-FS by increasing SST feedback periodicity until C–
18days.

The Hovmöller diagrams in Fig. 2a–i depict the evolution of 10° N–10° S

averaged precipitation and U850 anomalies on intraseasonal timescales, represented
by lagged correlation coefficients between precipitation averaged over 10° S–5° N,
75–100° E. In GPCP/ERA5, there is observed eastward propagation of precipitation
and U850 from the eastern IO to the dateline, with precipitation leading U850 by
approximately a quarter of a cycle. The propagation speed of the 30–80-day filtered
U850 anomaly is 5 m s$^{-1}$ (Fig. 2a). However, the A–CTL simulations exhibit
westward-propagating signals over the IO and weak, slow eastward propagation over
the MC and WP (Fig. 2b), which is also reflected in the W-FS shown in Fig. 1b,
indicating enhanced westward propagation in wavenumber 2. The Hovmöller
diagrams of the high-frequency and low-frequency experiments (Fig. 2c–h) display
the key eastward propagation characteristics of both precipitation and U850, as well
as the phase relationship between them, except for C–30days. The simulated
correlations between precipitation and U850 anomalies in the experiments are
generally weaker compared to GPCP and ERA5, particularly when crossing the MC
into the WP. Fig. 2b and Fig. 2c–h highlight the contrast, indicating that coupling a 1-
D TKE ocean model can significantly enhance an AGCM ability to simulate key
characteristics of the MJO, such as amplitude, propagation direction and speed, and
the phase relationship between precipitation and circulation. When crossing the MC,
the Hovmöller diagram of C–3days precipitation exhibits a substantial weakening
compared to other high-frequency and low-frequency experiments (except for C–
30days). This weakening is accompanied by a weaker easterly zonal wind in the MC.
This finding is consistent with the results from the W-FS in Fig. 1e. Neither
precipitation nor U850 exhibit clear eastward propagation characteristics over C–
30days. Further detailed discussions on this topic will be presented in the subsequent
chapter.

We conducted a cross-spectral analysis to examine the phase lag and coherence

between the tropical circulation and convection. Figures 3a–i illustrate the symmetric
part of OLR and U850 for NOAA/ERA5 data, A–CTL, C–1day, C–3days, C–6days,
C–12days, C–18days, and C–30days, respectively. The MJO band exhibits a high
degree of coherence, indicating a strong correlation between NOAA MJO-related
OLR signal and wavenumbers 1–3 (Fig. 3a). The simulated phase lag in the 30–80-
day band is approximately 90°, consistent with previous studies (Ren et al., 2019;
Wheeler and Kiladis 1999). All model experiments show significant coherence within
wavenumber 3 in the MJO band, with a phase lag similar to NOAA/ERA5 data.
However, A–CTL at wavenumber 1 only exhibits half of the observed coherency
peaks, and the coherence at wavenumbers 2–3 for the 30–80-day period is weaker
compared to NOAA/ERA5 data. The experiments C–CTL, C–1day, C–3days, C–
6days, C–12days, and C–18days exhibit similar coherency peaks to NOAA/ERA5 at
wavenumber 1. Additionally, as the SST feedback periodicity increases, the
experiments between C–12days and C–30days simulate unrealistic coherency over
wavenumber 9 in the MJO band (Fig. 3g–i).

The 20–100-day filtered precipitation anomalies (shaded) and SST anomalies

(contour) were averaged over the 10° S–10° N region (Fig. 4a–i). Phase-longitude
diagrams were used to analyze the relationship between precipitation and SST



fluctuations and to establish the connection between air–sea coupling and convection.
Except for C–30days, both GPCP/OISST and the coupled experiments clearly showed
the eastward propagation of enhanced convection with positive SST anomalies (Fig.
4a and 4c–i). The amplitude of SST increases in low-frequency experiments, as
indicated in Table 1 and Fig. 4f–h, resulting in precipitation anomalies lagging by
approximately 2–3 phases than SST, particularly when crossing the MC. Liang et al.
(2018) indicated SST leading precipitation by 10 days implies air–sea interactions at
the intraseasonal timescale during MJO events, with SST playing a crucial role in
modulating the MJO's intensity and propagation. The A–CTL simulations exhibited
weak SST anomalies and stationary precipitation when using the monthly average
SST interpolation from OISST. In contrast, the C–30days experiment showed
unrealistic SST and precipitation variability. Overall, eastward propagation of the
MJO is not favored by either minimal or large SST fluctuations (Fig. 4b and 4i). By
comparing the coupled experiments with the aforementioned simulations, it became
evident that air–sea interaction plays a crucial role in facilitating eastward
propagation. Fu et al. (2017) found that a robust intraseasonal SST anomaly is
associated with successive MJO events and supports the propagation of MJOs, as
supported by NOAA OLR and TRMM precipitation. This study highlights the
significant improvement in eastward propagation simulations achieved by
incorporating the air–sea interaction process into the model with distinct high-
frequency and low-frequency experiments, even with a simple 1-D ocean model like
SIT.

**3.2.2 Vertical structures of the MJO in the atmosphere**

Air–sea interaction plays a significant role in influencing atmospheric moisture

and convection associated with the MJO (Savarin and Chen, 2022). During periods of



convective suppression, the surface air temperature generally tracks the SST closely
(de Szoeke et al., 2014). A warmer upper ocean enhances low-level atmospheric
convergence, leading to increased low-level moisture and preconditioning that
facilitate eastward propagation and deep convection (DeMott et al., 2014). Hovmöller
diagrams in Fig. 5a–i illustrate the relationship between air temperature anomalies
(contoured, in K) and the vertically tilting structure of specific humidity (shading, in g
$kg^{-1}$) from the surface to the upper troposphere (200 hPa) over the 10° S–10° N and
120–150° E regions. Positive air temperature anomalies lead positive specific
humidity anomalies by approximately 2–3 phases, with the maximum specific
humidity occurring between 700–500 hPa. In ERA5 and the coupled experiments
(excluding C–30days), there are two relatively high values of air temperature at 300
and 700 hPa, respectively. However, in A–CTL, the maximum specific humidity
anomaly occurs at 700 hPa, and there is a vertically stationary structure in specific
humidity anomaly and an opposite tilting in air temperature (Fig. 5b). A–CTL also
exhibits a decrease in low-level moisture anomaly due to negative air temperature
anomalies below 700 hPa. C–30days, on the other hand, shows an unrealistic vertical
tilting structure in both specific humidity and air temperature anomalies (Fig. 5i).

According to the WISHE-moisture mode theory (Fuchs and Raymond, 2017), the

combination of mean easterly zonal winds and moisture plays a role in the
propagation and destabilization of the MJO. East of the convective MJO, enhanced
easterly winds induce atmospheric destabilization and moistening, leading to the
propagation of the MJO (Sentić et al., 2020). Figure 6 displays the averaged p-
vertical velocity anomaly (OMEGA, Pa $s^{-1}$, shaded) and zonal wind anomaly (m $s^{-1}$,
contour, interval 0.5) between phase 3 and phase 4 over the 15° N–15° S region. We
specifically selected the phase between 3 and 4 to examine the period leading up to
the MJO convection crossing the MC. Prior to the onset of the MJO in this phase,



there is typically a buildup of convection over the land areas of the MC, which
encompass countries such as Indonesia, Malaysia, and the Philippines. This land
convection acts as a precursor to the MJO as it creates favorable conditions and sets
the stage for the subsequent development of organized atmospheric disturbances. This
can be observed in the low-level ascending OMEGA shown in Figure 6a, specifically
between 120–150° E. The land convection over the MC is driven by a combination of
factors, including the local geography, land-ocean temperature contrasts, and large-
scale atmospheric conditions. The complex topography and the presence of extensive
water bodies surrounding the MC provide favorable conditions for the uplift of moist
air, which leads to the formation of local convection. Additionally, the temperature
differences between the warm ocean waters and the relatively cooler land surfaces
contribute to the instability and uplift of air masses.

In C–CTL, there is an enhanced easterly wind anomaly between 120° E and 180°

E at 800–600 hPa (Fig. 6c). The stronger easterly winds, coupled with radiative
heating, such as net downwelling surface solar radiation, lead to warmer upper ocean
temperatures (not shown). This heat stored in the upper ocean influences surface
fluxes and drives convection in the atmosphere (de Szoeke et al., 2014; Hsu et al.,
2019). In the western IO and MC region (Fig. 6c–h), there is a spatial distribution of
negative OMEGA (ascending motion) anomalies during phase 3–4, accompanied by
westerly wind anomalies to the west of MJO convection below 500 hPa in the coupled
experiments (except C–30days). In A–CTL during phase 3–4, negative OMEGA
anomalies are observed both east and unrealistically west of the MC (Fig. 6b).
Generally, the low-frequency experiments exhibit stronger negative OMEGA,
westerly wind anomalies and land convection compared to the high-frequency
experiments, except for C–30days. In the case of C–30days, deep convection in the



IO, MC, and WP regions is weakened as local convection occurs randomly during
phase 3–4 (Fig. 6i).

**3.2.3 The vertical structure of the ocean responds to the MJO**

Understanding the interaction between the atmosphere and upper ocean is

essential for studying the MJO, particularly the upper ocean's response to strong
atmospheric forcing. Accurate representation of air–sea interactions, including
momentum and heat fluxes associated with the MJO, is crucial for capturing the MJO
in coupled model simulations, as emphasized by Hong et al. (2017). It is crucial to
properly couple an ocean model that incorporates MJO air–sea interaction for a
comprehensive understanding. Such coupling can lead to warmer or cooler surface
oceans and shallower or deeper mixed layer depths before or after MJO convection in
the tropics, resulting from improved vertical resolution in the upper ocean (Tseng et
al., 2015; Lan et al., 2022). These oceanic changes, in turn, induce atmospheric
responses through the ocean feedback process. In this study, we employ a SIT model
coupled with CAM5 to investigate the frequency of air–sea coupling and its impact on
MJO simulation. This ocean model incorporates a high vertical resolution that
captures important features such as the cool skin layer and diurnal warm layer, as well
as the gradient of temperature in the upper ocean. In the OISST, high-frequency, and
low-frequency experiments, the strongest phase anomalies of maximum SST occur
between phases 2 and 3 (as shown in Table 1), particularly in the region of (110–130 °
E, 5–15° S), where SST variations associated with the MJO are most prominent.
Figure 7 illustrates the average oceanic temperature between 0- and 60-meters depth
during phase 2–3, filtered for the 20–100-day period, represented by shaded and
contour plots with an interval of 0.03 Kelvin. In the high-frequency experiments, the
upper oceanic temperatures exhibit warming patterns within 30 meters depth at 100–





473 140° E, while cooling is observed near the dateline (as shown in Fig. 7a–c). During

474 phase 2–3, as the MJO convection progresses into the IO (60–90° E), it interacts with

475 the ocean surface, leading to a cooling effect in the upper ocean in the C–CTL

476 experiment (Fig. 7a), which is more pronounced compared to the C–3days experiment

477 (Fig. 7c), characterized by stronger interseasonal MJO variability. In the low-

478 frequency experiments, the spatial distribution of warmer upper ocean temperatures is

479 more extensive than in the high-frequency experiments, spanning from the MC to the

480 WP. Additionally, the vertical temperature gradient in this region is greater in the low-

481 frequency experiments compared to the high-frequency experiments. However, in the

482 Indo-Pacific region, the C–30days experiment exhibits an unrealistic spatial

483 distribution of oceanic temperature anomalies, with small areas of both positive and

484 negative anomalous temperature fluctuations (Fig. 7g).

486 **4. Discussion**

487 **4.1 Empirical Orthogonal Function (EOF) analysis**

488  Empirical orthogonal functions (EOF) analysis helps identify spatial patterns and

489 their associated temporal variations, providing insights into the underlying dynamics

490 and relationships within the dataset. The first EOF mode captures the largest fraction

491 of variance in the data, while subsequent modes capture progressively smaller

492 amounts of variance and represent additional patterns. This study builds upon

493 previous research utilizing uncoupling simulations (e.g., DeMott et al., 2014; Stan,

494 2018) and investigates the influence of interseasonal SST feedback on the MJO by

495 incorporating real air–sea coupling. Figure 8 illustrates the near-equatorial variance

496 explained by the Real-time Multivariate MJO series 1 (RMM1) and series 2 (RMM2),

497 which represent the combined variance explained by the first two empirical

498 orthogonal functions (EOF1 and EOF2) according to Wheeler and Hendon (2004).

These RMM1 and RMM2 components have proven useful for estimating and
characterizing the MJO and its variability. The coupled experiments, when combined
with RMM1 and RMM2, demonstrate improved MJO performance compared to A–
CTL, with the exception of C–30days. The extension of interseasonal SST feedback
periodicity corresponds to an increase in RMM1+RMM2. Generally, the
RMM1+RMM2 percentages are larger for the low-frequency experiments compared
to the high-frequency experiments, except for C–30days. Among the high-frequency
experiments, C–3days shows a slight degradation in RMM1+RMM2, consistent with
the observed MJO-related results such as the W-FS for 850-hPa zonal wind (Fig. 1e),
the Hovmöller diagram of precipitation crossing the MC (Fig. 2e), the OLR power
spectrum at a zonal wavenumber-frequency wavelength (Fig. 3e), the maximum
phase-vertical Hovmöller diagram of 20–100-day specific humidity between 700–500
hPa (Fig. 5e), and the cooling effects in the upper ocean (Fig. 7c). However, C–
30days exhibits lower skill in terms of RMM1+RMM2 compared to the other coupled
simulations due to the presence of excessive local convection and weak large-scale
circulation, which is reflected in the unrealistic spatial distribution of oceanic
temperature.

**4.2 The dynamic lead–lag relationship of intraseasonal variability**

The lead–lag relationship refers to a situation where one variable (leading) is

cross-correlated with the values of another variable (lagging) in subsequent phases,
particularly in the case of SST fluctuations and MJO-related atmospheric variations
between phase 1 and 8 within the domain of 110–130° E and 5–15° S (Fig. 9). The
analyzed variables consist of 20–100-day filtered latent heat flux (LHF, indicated by
green shading), OLR (indicated by a yellow bar chart), net surface solar radiation
(FSNS, indicated by an orange bar chart), U850 (indicated by a purple bar chart), 30-



meter depth oceanic temperature (30-m T multiplied by 100, indicated by a black
line), and SST (multiplied by 10, indicated by an orange line) which positive values
are represented by an upward direction in LHF and FSNS. The graphical
representation of variables marked with "(L)" employs the left y-axis, while variables
marked with "(R)" utilize the right y-axis.
The decrease in LHF, which indicates a reduction in heat loss from the ocean,
and the negative FSNS, indicating that solar radiation is heating the ocean, coincide
with easterly zonal winds that contribute to positive SST anomaly in ERA5 (Fig. 9a).
This lead–lag relationship depicts the changes in LHF, FSNS, OLR, U850 and SST
which positive SST anomaly prior to the MJO convection period emphasizing the
interconnectedness of oceanic heat fluxes, solar radiation, and atmospheric circulation
patterns. As the MJO convection progresses through the region (110–130° E and 5–
15° S), several changes in atmospheric and oceanic variables occur. These changes
include a shift in OLR from positive to negative values, a decrease in SST, a transition
to westerly winds, and an increase in positive FSNS and LHF (Fig. 9a). With the
exception of experiments of A–CTL and C–30days, both the high-frequency and low-
frequency SST feedback experiments exhibit similar simulation of lead–lag
relationships when compared to ERA5 (Fig. 9c–h). It is worth noting that in
experiments C–CTL, C–1day, C–3days, and C–6days, the variations in LHF are
underestimated. Conversely, in experiment C–18days, the variations in LHF are
overestimated. In experiment C–12days, the variations in LHF are similar to the
expected values. The magnitude of SST fluctuations is directly related to the
variations in LHF, FSNS, OLR, and U850 in the lead–lag relationship. In ERA5,
phase 2 corresponds to the occurrence of the maximum positive SST anomaly within
the domain of 110–130° E and 5–15° S, while phase 7 corresponds to the occurrence
of the most negative SST anomaly. When comparing the high-frequency and low-



frequency SST feedback experiments to ERA5, except for experiments A–CTL and
C–30days, the maximum positive SST anomaly is consistently delayed by one phase.

Additionally, the occurrence of the most negative SST anomaly aligns with the

same phase in both types of experiments. The maximum positive anomaly in the 30-m
T is delayed by one phase compared to the SST, indicating the transfer of heat from
the ocean surface into the upper ocean progressively. Similarly, the occurrence of the
most negative 30-m T anomaly is also delayed by one phase compared to SST,
revealing the buffering role of the upper ocean when the MJO convection extracts
heat (energy) from the ocean (Fig. 9c–i). In the A–CTL experiment, which utilizes
monthly OISST data, the SST anomalies are relatively small. This is reflected in the
weak anomalies observed in OLR and FSNS (Fig. 9b). On the other hand, in the C–
30days experiment, there is a misalignment in the lead–lag relationship, and the OLR
and FSNS anomalies are also weak (Fig. 9i).

**4.3 The extreme frequency of oceanic feedback can sustain MJO propagation**

In previous studies, it has been observed that most models incorporate both

coupled and uncoupled simulations. DeMott et al. (2014) specifically noted that in
uncoupled experiments, SPCAM3 exhibited strong eastward propagation for 5-day
running means, but relatively weaker propagation for monthly means. This raises the
question of how much SST feedback periodicity is necessary to maintain robust
eastward propagation in coupled experiments. This section aims to discuss this topic
and explore strategies for achieving robust eastward propagation. It is observed that
the aforementioned criteria are met with increased feedback periodicity for SST until
the C–30days experiment. SST feedback periodicity, characterized by SST-forced
atmospheric variability, exhibits notable differences between coupled and uncoupled
experiments. In uncoupled experiments (A–CTL), the SST lacks responsiveness to



atmospheric changes, leading to unrealistic intraseasonal variability in atmospheric
circulation. Spatially, Through air–sea interaction, most of the coupled experiments
showed improved MJO simulation with realistic strength and eastward propagation
speeds (e.g., C–CTL, C–1day, C–3days, C–6days, C–12days, and C–18days), where
higher MJO variance was associated with increased SST feedback periodicity.

Generally, C–18days exhibited an overestimation of intraseasonal variability

while maintaining eastward propagation of the MJO. Figure 10 highlights
considerable differences in the simulation of robust (disordered) MJOs at phase 4
between C–18days and C–30days. In C–18days, negative OLR anomalies are
widespread across the MC and extend to the WP near the equator in the northern
hemisphere (Fig. 10b). Concurrently, U200 exhibits divergence patterns that coincide
with the negative OLR anomaly. Negative OLR anomalies are indicative of the
presence of deep convection. In the C–CTL experiment, the spatial distribution of
negative OLR overlaps with positive net surface heat flux and solar radiation
anomalies, indicating heat loss from the ocean to the atmosphere (Fig. 10g). Notably,
in C–18days, there is irregular heat flux loss from the surface ocean near the equator
in the western Pacific (Fig. 10h), which is not observed in C–CTL.

Furthermore, in the C–18days experiment, there is a notable 75% increase in

solar radiation anomalies (with upward direction indicating positive values), resulting
in reduced solar radiation reaching the ocean surface in the southeastern IO when
compared to C–CTL. Positive anomalies in LHF (Fig. 10e) are predominantly
observed within and west of the convective region, coinciding with westerly
winds and a cooling of SST (Fig. 10k). Liang et al. (2018) investigated the variability
of the heat fluxes, a major contributor of the intraseasonal SST variability. In C–CTL
at phase 4, relatively weak westerly winds and latent heat flux are observed in the IO
(Fig. 10d). Upon the passage of deep convection across the MC, the IO experiences



intensified westerly winds and latent heat flux anomalies (not shown). These positive
latent heat flux anomalies, resulting from ocean evaporation, contribute negative SST
anomalies and provide a negative feedback in the atmosphere. Wu and Kirtman
(2005) suggest that through air–sea coupling, SST-forced atmospheric changes in
surface winds and heat fluxes exert a strong negative feedback on SSTs in the Indo-
Pacific region. Jayakumar et al. (2011) conducted a series of experiments using an
ocean general circulation model to investigate the individual contributions of different
processes. Their findings during the period of 1997–2006 reveal that wind stress
accounted for approximately 20% of the intraseasonal SST variability in the IO
region, while heat fluxes made up about 70% of the variability. Among the heat flux
components, shortwave radiation exerted the most significant influence, contributing
75%, while the remaining 25% was attributed to other flux components. Gao et al.
(2020a) corroborate that the temporal variations in SST anomaly are primarily driven
by shortwave radiative heating, and LHF playing a secondary role.

In both C–CTL and C–18days simulations, there is evidence of a negative

feedback in the lag–lead relationships among SST, surface winds, rainfall anomalies,
and heat fluxes (Fig. 9c and 9h), which supports the findings of previous studies (Wu
and Kirtman, 2005; Jayakumar et al., 2011; Gao et al., 2020a). With increased
feedback periodicity of SST in C–CTL and C–18days, the ocean continues to receive
atmospheric forcing, but the feedback response is delayed, leading to the
accumulation of energy (temperature) in the upper ocean, as seen in the SST
distribution in the WP (Fig. 10k). In C–30days, SST exhibits a perturbed
unrealistically spatial distribution (Fig. 10l) driven by plus-minus latent heat flux and
10m wind anomalies (Fig. 10f), net surface heat flux, and solar radiation (Fig. 10i).
Consequently, these perturbed SST plus-minus patterns trigger numerous local





convections among the IO, MC, and WP and does not manifest as organized the large-
scale circulation.

**4.4 The moist static energy (MSE) analysis**
A budget analysis of MSE is used to investigate the underlying mechanisms
driving the onset and eastward propagation of the MJO event. Analyzing the MSE
budget provides valuable insights into the physical processes and feedback
mechanisms influencing the behavior of the MJO, including vertical MSE advection,
zonal MSE advection, meridional MSE advection, surface heat fluxes, atmospheric
radiative term, and residual components.

**4.4.1 Preconditioning phase**
Analysis of the column-integrated MSE budget has revealed that both vertical
and horizontal MSE advection contribute to the east-west asymmetry of MSE
tendency, thereby facilitating the eastward propagation of the MJO (Wang and Li,
2020). Figure 11 illustrates the physical processes associated with each term
contributing to the column-integrated MSE tendency (<dmdt>) in Eq. (1), as outlined
in Tseng et al. (2022), preceding deep convection over the MC area (10° S–0° N/S,
120–150° E) during phase 2 over the ERA5 and model simulations. The MJO
convection in the eastern Indian Ocean at phase 2, the column-integrated vertical
advection (-<wdmdp>) over the MC area takes a dominant role in the MSE budget,
while horizontal MSE advection (-<vdm>) plays a secondary role. These findings,
along with significant compensation from longwave radiation, were identified by
Wang and Li (2020) and Tseng et al. (2022). Generally, the -<wdmdp> accounts for
approximately 72–86% of ERA5, except for A–CTL and C–30days, while the -
<vdm> increases from 40% to 80% of ERA5 due to the heightened feedback





periodicity of SST (Fig. 11). Those results indicate that all model simulations exhibit
weaker 20–100-day filtered MSE advection anomalies prior to the eastward
propagation of the MJO over the MC compared to ERA5. Similarly, the precipitation
results in Fig. 4c–h demonstrate the same trend. Moreover, the LH term exhibits an
opposite trend to the -<vdm> term due to the increased feedback periodicity of SST,
while SH and shortwave radiation fluxes (<SW>) contribute less to the negative MSE
tendency in both ERA5 and model simulations. These findings indicate that, in the
early phase, the negative contribution primarily stems from the LH and longwave
radiation fluxes (<LW>) term. Tseng et al. (2022) identify the negative LH bias as a
key factor in enhancing the leading MSE tendency during MJO preconditioning
phases. In general, coupling enhances the budget simulation by increasing the positive
contribution of vertical and horizontal advection and the negative contribution of LH
and <LW> in MSE tendency, primarily due to the intensified feedback periodicity of
SST during the initial phase of the MJO. Although the <dmdt> term can be further
decomposed into variations of <wdmdp>, <vdm>, and LH in model simulations, the
total column-integrated MSE tendency does not exhibit a clear difference in response
to the increasing feedback periodicity of SST experiments during the initial phase of
the MJO.

**4.4.2 Phase of strongest convection across the MC**

We compared the spatial distribution of 20–100-day <dmdt> (shading),

precipitation (contours), and 850-hPa wind (vectors) during phase 5, which represents
the period of strongest convection across the MC (Fig. 12). In ERA5, the main
convection is accompanied by positive precipitation anomalies and low-level
convergence in the 850-hPa wind across the MC extending into the WP (Fig. 12a). A
positive MSE tendency, peaking near 15° N and 15° S, is observed to the east of the





MJO convection located near the Equator. Conversely, a negative integrated MSE
tendency is observed to the west of the MJO convection accompanied by negative
precipitation anomalies to the west of this region. The meridionally confined structure
near the Equator exhibits characteristics indicative of an equatorial Kelvin wave
propagated toward the east as fundamental dynamics of the MJO. With the exception
of A–CTL and C–30days, the model simulations display a similar structure to ERA5
in terms of the 20–100-day filtered <dmdt>, precipitation, and 850-hPa wind vectors
(Fig. 12c–h). C–CTL exhibits relatively weak precipitation anomalies in the MC and
weak westerly winds in the IO until C–6days, where robust precipitation and low-
level convergence in the 850-hPa wind occur in response to the feedback periodicity
of SST increasing. On the contrary, A–CTL exhibits abnormal positive precipitation
anomalies distributed over the western IO, while localized maximum of <dmdt>
occur near 15° N (Fig. 12b). In contrast, C–30days displays plus-minus precipitation
anomalies near the Equator, consequently disrupting the spatial distribution of the
<dmdt> relative to MJO convection (Fig. 12i).

To quantify the impact of SST feedback periodicity on atmospheric

intraseasonal variability in the tropics, we adopt the approach of Tseng et al. (2022)
and Jiang et al. (2018) to project all MSE terms onto the 20–100-day filtered ERA5
<dmdt> (Fig. 12a) during phase 5. The MC has been frequently identified as a barrier
to the eastward propagation of the MJO, as noted by Li et al. (2020b). Additionally, a
considerable proportion, approximately 30–50%, of the MJO experiences stalling
over the MC, as reported by Zhang and Han (2020). To mitigate the influence of
weaker MJO events that dissipate prior to reaching the MC, our focus is specifically
on phase 5 of the MJO. Figure 13(a) illustrates the determination of the contribution
of each component of the MSE tendency during phase 5 by projecting the spatial
pattern of each MSE budget term over the MC region (20° S–20° N, 90–210° E),



where $F_s$ is total surface fluxes including sensible and latent heat fluxes, and $Q_r$ is
vertically integrated radiative (short-wave and long-wave) heat fluxes. The dominant
contribution of horizontal advection to the MSE tendency (Fig. 13a) are simulated
well in both the high-frequency and low-frequency SST feedback experiments, but
not in the A–CTL simulation. The -<vdm> term increases in response to the
increasing feedback periodicity of SST, resulting from stronger low-level
convergence, which enhances MJO convection. Vertical advection -<wdmdp> is not
the dominant term over the MC region (20° S–20° N, 90–210° E) in both ERA5 and
model simulations during phase 5. Furthermore, Fs and Qr make a minor contribution
to the MSE tendency, with the sensible and latent heat fluxes exhibiting a tendency
towards greater recessive behavior in response to the increasing feedback periodicity
of SST.
The total horizontal MSE advection is further decomposed into its zonal (-
<udmdx>) and meridional zonal (-<vdmdy>) components for high-frequency SST
feedback experiments (C–CTL, A–CTL, C–1day, and C–3days) and low-frequency
SST feedback experiments (C–6days, C–12days, C–18days, and C–30days) in order
to examine their individual effects (Fig. 13b–c). Both components contribute
positively, but the -<vdmdy> exhibits a larger amplitude, consistent with findings by
Tseng et al. (2022) during phase 4. The -<vdmdy> of high-frequency SST feedback
experiments (C–CTL, C–1day, and C–3days) closely resemble ERA5 in terms of the
projected magnitude. Comparatively, the -<vdmdy> term in low-frequency SST
feedback experiments (C–6days, C–12days, C–18days, and C–30days) exhibits a
more positive contribution than in high-frequency SST experiments, leading to a
dominant contribution to the increase in -<vdm> and <dmdt>.
We generated a spatial representation of the 20–100-day column-integrated
vertical MSE advection ($J\,kg^{-1}\,s^{-1}$, represented by shading), column-integrated



horizontal MSE advection ($J\,kg^{-1}\,s^{-1}$, shown as contours with an interval of 6.0), and
200-hPa wind (green vectors) relative to a reference vector ($3\ m\ s^{-1}$) during phase 5
(Fig. 14). This figure complements the information provided by the bar chart in Fig.
13a. In ERA5, the wind divergence at 200 hPa during phase 5 (Fig. 14a), overlaid
with the 850-hPa convergence (Fig. 12a), indicates a vertically tilting structure of
zonal wind anomalies. Except for A–CTL and C–30days, the model simulations
exhibit a similar structure to ERA5 in terms of low-level convergence and high-level
divergence. In ERA5, the negative -<wdmdp> and -<vdm> anomalies (Fig. 14a) are
observed to the west of the MJO convection, which is characterized by positive
precipitation anomalies (Fig. 12a). The spatial distribution of the negative -<vdm>
anomaly (dashed-red contours) extends from the IO to the MC, exhibiting a pattern
similar to <dmdt> with enhanced anomalies. This results in the projection of the
spatial pattern of the -<vdm> term being greater than 1. The positive -<wdmdp>
anomaly (shading) is located in the western IO and east of the dateline, which results
in a spatial distribution unlike that of <dmdt> comparatively. This difference reduces
the projection of the spatial pattern of -<vdm> to a value lower than 1. On the
contrary, in the A–CTL experiment, the positive -<vdm> anomaly (solid-blue
contours) exhibits a spatial distribution near 120° E (Fig. 14b), while the negative -
<vdm> anomaly (dashed-red contours) is distributed on both the positive left and
right sides. Although the negative -<vdm> anomaly in high-frequency SST feedback
experiments (C–CTL, C–1day, and C–3days) underestimates that of ERA5 (Fig. 14c–
e), the spatial distribution remains similar to ERA5 due to an approximately 80%
projection of -<vdm> compared to ERA5. The low-frequency SST feedback
experiments (C–6days, C–12days, and C–18days) yield greater -<vdm> anomalies
(Fig. 14f–h) compared to ERA5, with projection values greater than 1. We noticed
that, in the low-frequency SST feedback experiments, although the anomalies of -



<wdmdp> intensify, the spatial distribution of those shift eastward, leading to a
decrease in projection values.

**5. Conclusions**
This study builds upon the work of Lan et al. (2022) and Tseng et al. (2022) by
coupling a high-resolution 1-D TKE ocean model (the SIT model) with the CAM5,
specifically the CAM5–SIT configuration, to investigate the extreme effects of
interseasonal SST feedback on the MJO. We introduced asymmetric exchange
frequencies between the atmosphere and the ocean, ensuring bidirectional interaction
at each timestep within the experimental periodicity by fixing the SST value in the
Coupler. This allowed us to create various intraseasonal SST feedback atmospheric
experiments, including intervals of 30 minutes, 1, 3, 6, 12, 18, and 30 days.
Systematic sensitivity experiments were conducted to divide into two groups: those
feedback periodicity within a phase (high-frequency SST) and those beyond a phase
(low-frequency SST).
The aim is to assess the scientific reproducibility and consistency of the findings
across different SST feedback cycles in the field of modeling science. With the
exception of the C–30days experiment, both the high-frequency (C–CTL, C–1day,
and C–3days) and low-frequency (C–6days, C–12days, C–18days) experiments
demonstrate realistic simulations of various aspects of the MJO when compared to
ERA5. These aspects include intraseasonal periodicity (as shown in Fig. 1), eastward
propagation (as observed in Fig. 2 and 4), coherence in the low-frequency band (as
depicted in Fig. 3), tilting vertical structure (evident in Fig. 5, 12, and 14 for zonal
wind), intraseasonal SST (as summarized in Table 2) and oceanic temperature
variances (as shown in Fig. 7), the lead–lag relationship of intraseasonal variability
(as characterized in Fig. 9), phase 2 column-integrated MSE tendency terms





(including decomposition items) (illustrated in Fig. 11), and the projection of all MSE
terms onto the ERA5 column-integrated MSE tendency during phase 5 (depicted in
Fig. 13).

The lead–lag relationship provides a visual representation of the variations in

20–100-day filtered LHF, FSNS, OLR, U850 and SST, while positive SST leading up
to the onset of the MJO convection (Fig. 9). This relationship highlights the
interconnected nature of oceanic heat fluxes, solar radiation, and atmospheric
circulation patterns, underscoring their mutual influence and interplay. Table 3
provides a comprehensive overview of several variables during the boreal winter,
including the average values of 20–100-day filtered OLR, LHF, FSNS, U850,
<dmdt>, -<wdmdp>, and -<vdm>. These variables are categorized based on the states
of SST warming and cooling. The categorization is performed over two specific
domains: (110–130° E, 5–15° S), as referenced in Fig. 9, and (120–150° E, 0–10° S)
marked as background gray, as referenced in Fig. 11. We highlight the characteristics
of the MJO-related atmosphere with red letters, which correspond closely to the
values in ERA5. In synthesizing the findings from Arnold et al. (2013) regarding the
high SST enhances MJO simulation, the improved MJO simulation through
intraseasonal SST variability by Liang et al. (2018), the information provided in
Tables 2 and 3, and the corresponding figures, it becomes evident that the high-
frequency (low-frequency) SST experiments tended to underestimate (overestimate)
the MJO simulation. Notably, the experiment C–6days demonstrated the closest
similarity to ERA5 in terms of MJO simulation.

Among the high-frequency experiments, C–3days shows a less close

resemblance to the observed MJO characteristics. The result of the C–3days
experiment is consistent with Stan (2018), as the absence of 1–5-day variability in
SST promotes the amplification of westward power associated with Rossby waves. In



addition, the C–1day experiment confirms the scientific reproducibility of Hagos et al.
(2016) and Lan et al. (2022) that demonstrates that the removal of the diurnal cycle
enhances the MJO.

The increasing feedback periodicity of SST in low-frequency experiments leads

to the accumulation of short-wave and long-wave radiations and surface heat fluxes
from the atmosphere, resulting in an increase in the upper oceanic temperature and its
variances (Table 2). SST variances that are higher than OISST contribute to
robust/overestimated simulations of the MJO (as observed in Fig. 1–14 and Table 3).
In contrast, the C–30days experiment exhibits variances with both positive and
negative anomalies in precipitation (Fig. 4 and 10), oceanic temperature (Fig. 7), net
surface heat fluxes (Fig. 10), and column-integrated vertical and horizontal MSE
advection (Fig. 14). These anomalies have an unrealistically spatial distribution and
an unrealistic vertical tilting structure in both specific humidity and air temperature
anomalies (Fig. 5i) over the Indo-Pacific region. As a result, local convection appears
randomly among the IO, MC, and WP, and does not manifest as organized MJO
convection.

Finally, in Fig. 15, the interseasonal SST feedback experiments on MJO are

depicted schematically. These experiments include the uncoupled model (A–CTL),
high-frequency SST experiments (C–CTL, C–1day, and C–3days), low-frequency
SST experiments (C–6days, C–12days, C–18days), and disorganized convection and
circulation (C–30days) which figure concept is based on DeMott et al. (2014) in Fig.
11. In the absence of interseasonal SST variability, the uncoupled A–CTL disrupts the
eastward propagation of the MJO, leading to weakened or fragmented MJO activity as
shown in Fig. 15a. On the other hand, the high-frequency SST experiments generally
capture the characteristics of the MJO. The time-varying SSTs in the coupled
simulation provide a certain level of organization and sufficient surface fluxes, which



facilitate the development of MJO circulations, as illustrated in Fig. 15b. Moreover, in
the coupled model, the presence of land convection over the MC ahead of the MJO
convection (Fig. 6) contributes to the instability and uplift of moist air masses.
Conventionally, the MJO has been regarded as a tropical atmospheric variability,
given that its existence is primarily attributed to the interplay between organized
convection and large-scale circulations. This dynamic process plays a crucial role in
triggering the eastward propagation of the MJO. Furthermore, the low-frequency SST
experiments demonstrate robust simulations of the MJO. This can be attributed
comprehensively to the increased SST variances, accumulation of surface fluxes,
enhanced low-level convergence (Fig. 12) and high-level divergence (Fig. 14), as well
as horizontal MSE advection, as depicted in Fig. 15c. On the other hand, the C–
30days experiment simulates frequent, disorganized convection, as shown in Fig. 15d.
This experiment exhibits both positive and negative anomalies in precipitation, SST,
surface heat fluxes, and vertical and horizontal MSE advection, which fail to generate
the expected circulation anomalies.

*Code and data availability.* The model code of CAM5–SIT is available at
https://doi.org/10.5281/zenodo.5510795. Input data of CAM5–SIT using the
climatological Hadley Centre Sea Ice and Sea Surface Temperature dataset and
GODAS data forcing, including 30-year numerical experiments, are available at
https://doi.org/10.5281/zenodo.5510795.

*Author contributions.* HHH is the initiator and the primary investigator of the
Taiwan Earth System Model project. YYL is the CAM5–SIT model developer and
writes the majority part of the paper. WLT assistṣ in MSE analysis.



*Competing interests*. The authors declare that they have no conflict of interest.

*Acknowledgements.* The contribution from YYL, HHH, and WLT to this study is
supported by Ministry of Science and Technology of Taiwan under contracts MOST
110-2123-M-001-003, MOST 110-2811-M-001-603, MOST 109-2811-M-001-624
and MOST108-2811-M-001-643. Our deepest gratitude goes to the editors and
anonymous reviewers for their careful work and thoughtful suggestions that have
helped improve this paper substantially. We sincerely thank the National Center for
Atmospheric Research and their Atmosphere Model Working Group (AMWG) for
release CESM1.2.2. We thank the computational support from National Center for
High530 performance Computing of Taiwan. Thanks, ChatGPT for correcting the
English grammar.





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



Table 1. Two sets of marine feedback frequency with high-frequency
SST feedback (C–CTL, C–1day and C–3days) and low-frequency
SST feedback (C–6days, C–12days, C–18days and C–30days) under
SST sub-seasonal variability.

| subseasonal sets | high-frequency SST (< 6 days) | | | low-frequency SST (6-30 days) | | | |
|---|---|---|---|---|---|---|---|
| experiments | C–CTL | C–1day | C–3days | C–6days | C–12days | C–18days | C–30days |
| atmosphere to ocean frequency | | | | 48/day | | | |
| ocean to atmosphere Frequency | 48/day | 1/1day | 1/3days | 1/6days | 1/12days | 1/18days | 1/30days |






Table 2. The average DJF temperature difference between SST and 10m depth (
$\overline{\Delta T_{0-10m}}$) and 30m depth ($\overline{\Delta T_{0-30m}}$), and the boreal winter phase mean of 20–100-
day bandpass filter with max/mini SST and oceanic 10m depth temperature ($T_{10m}$) in
the area of (110–130° E, 5–15° S), with observation (OISST), AGCM (A–CTL), high-
frequency experiments (C–CTL, C–1day and C–3days) and low-frequency
experiments (C–6days, C–12days, C–18days and C–30days)

| (110–130° E, 5–15° S) | | obs. | AGCM | high-frequency | | | low-frequency | | | |
|---|---|---|---|---|---|---|---|---|---|---|
| experiments | | OI SST[1] | A–CTL[2] | C–CTL | C–1day | C–3days | C–6days | C–12days | C–18days | C–30days |
| **DJF seasonal mean** | SST | 302.2 ±0.96 | 302.2 ±0.77 | 300.8 ±0.76 | 301.2 ±0.76 | 301.2 ±0.75 | 301.2 ±0.75 | 301.4 ±0.75 | 301.6 ±0.80 | 302.7 ±1.71 |
| | $\overline{\Delta T_{0-10m}}$ | - | - | 0.1 ± 0.22 | 0.1 ± 0.22 | 0.1 ± 0.21 | 0.1 ± 0.23 | 0.2 ± 0.25 | 0.3 ± 0.32 | 1.0 ± 0.95 |
| | $\overline{\Delta T_{0-30m}}$ | - | - | 0.8 ± 0.79 | 0.7 ± 0.70 | 0.6 ± 0.69 | 0.8 ± 0.70 | 0.8 ± 0.70 | 1.0 ± 0.73 | 2.1 ± 1.54 |
| **phase's mean in boreal winter** | max SST (phase) | 0.21 (ph2) | 0.02 (ph2) | 0.24 (ph3) | 0.26 (ph3) | 0.22 (ph3) | 0.32 (ph3) | 0.36 (ph3) | 0.43 (ph3) | 0.62 (ph2) |
| | max $T_{10m}$ (phase) | - | - | 0.15 (ph4) | 0.17 (ph4) | 0.14 (ph3) | 0.19 (ph3) | 0.21 (ph3) | 0.26 (ph3) | 0.35 (ph2) |
| | mini SST (phase) | -0.21 (ph7) | -0.003 (ph8) | -0.17 (ph7) | -0.22 (ph7) | -0.19 (ph7) | -0.25 (ph7) | -0.28 (ph7) | -0.38 (ph7) | -0.60 (ph6) |
| | mini $T_{10m}$ (phase) | - | - | -0.11 (ph8) | -0.12 (ph7) | -0.11 (ph8) | -0.15 (ph7) | -0.17 (ph7) | -0.24 (ph7) | -0.33 (ph6) |

Note: [1] daily average data, [2] monthly average data.



Table 3. The average 20–100-day filtered outgoing longwave radiation (OLR), latent
heat flux (LHF), net surface solar radiation (FSNS), 850-hPa zonal wind (U850),
column-integrated MSE tendency (<dmdt>), column-integrated vertical MSE
advection (-<wdmdp>), and column-integrated horizontal MSE advection (-<vdm>)
during the boreal winter are categorized into SST warming and SST cooling states.
This categorization is performed over two domains, namely (110–130° E, 5–15° S)
and (120–150° E, 0–10° S), as mentioned in the note.

| | experiments | obs. ERA5/NOAA | AGCM A–CTL | high-frequency C–CTL | C–1day | C–3days | low-frequency C–6days | C–12days | C–18days | C–30days |
|---|---|---|---|---|---|---|---|---|---|---|
| **SST warming** | OLR[1] | 16.3 | 6.3 | 14.8 | 16.5 | 16.0 | 18.5 | 19.5 | 19.3 | 11.1 |
| | (phase) | (ph1) | (ph2) | (ph2) | (ph2) | (ph2) | (ph2) | (ph1) | (ph1) | (ph8) |
| | LHF[1] | -10.1 | -11.1 | -7.3 | -7.3 | -6.0 | -8.6 | -11.3 | -19.3 | -21.9 |
| | (phase) | (ph3) | (ph3) | (ph3) | (ph3) | (ph2) | (ph2) | (ph3) | (ph2) | (ph1) |
| | FSNS[1] | -15.7 | -8.9 | -15.7 | -17.9 | -15.9 | -19.5 | -18.6 | -16.8 | -9.5 |
| | (phase) | (ph1) | (ph2) | (ph2) | (ph2) | (ph2) | (ph2) | (ph1) | (ph2) | (ph1) |
| | U850[1] | -3.0 | -2.3 | -3.0 | -2.8 | -2.3 | -2.9 | -2.8 | -3.4 | -2.2 |
| | (phase) | (ph2) | (ph3) | (ph3) | (ph3) | (ph3) | (ph3) | (ph3) | (ph2) | (ph2) |
| | <dmdt>[2] | 10.7 | 9.1 | 8.2 | 8.2 | 5.6 | 7.9 | 8.1 | 7.0 | 4.1 |
| | (phase) | (ph3) | (ph3) | (ph3) | (ph3) | (ph3) | (ph3) | (ph2) | (ph2) | (ph3) |
| | -<wdmdp>[2] | 18.2 | 8.4 | 12.9 | 19.2 | 13.9 | 17.9 | 18.1 | 21.9 | 10.4 |
| | (phase) | (ph1) | (ph2) | (ph1) | (ph1) | (ph1) | (ph1) | (ph1) | (ph1) | (ph1) |
| | -<vdm>[2] | 11.5 | 5.3 | 7.9 | 7.9 | 4.4 | 8.4 | 9.0 | 9.1 | 14.0 |
| | (phase) | (ph2) | (ph3) | (ph3) | (ph3) | (ph3) | (ph3) | (ph3) | (ph2) | (ph1) |
| **SST cooling** | OLR[1] | -19.2 | -8.9 | -11.3 | -14.2 | -15.0 | -20.9 | -20.3 | -22.5 | -11.0 |
| | (phase) | (ph5) | (ph6) | (ph6) | (ph6) | (ph6) | (ph5) | (ph5) | (ph5) | (ph5) |
| | LHF[1] | 15.6 | 17.3 | 7.4 | 8.0 | 7.0 | 8.7 | 15.2 | 18.1 | 29.8 |
| | (phase) | (ph6) | (ph7) | (ph7) | (ph6) | (ph6) | (ph6) | (ph6) | (ph6) | (ph6) |
| | FSNS[1] | 19.7 | 10.5 | 11.6 | 16.6 | 16.1 | 21.9 | 19.1 | 21.6 | 10.4 |
| | (phase) | (ph5) | (ph5) | (ph6) | (ph6) | (ph6) | (ph5) | (ph5) | (ph5) | (ph5) |
| | U850[1] | 3.5 | 2.6 | 2.6 | 2.7 | 2.3 | 2.8 | 2.8 | 3.4 | 2.7 |
| | (phase) | (ph6) | (ph6) | (ph7) | (ph6) | (ph7) | (ph7) | (ph6) | (ph6) | (ph6) |
| | <dmdt>[2] | -10.6 | -7.5 | -9.0 | -7.9 | -6.0 | -9.0 | -8.2 | -8.9 | -3.8 |
| | (phase) | (ph6) | (ph7) | (ph7) | (ph6) | (ph6) | (ph6) | (ph6) | (ph6) | (ph7) |
| | -<wdmdp>[2] | -23.6 | -9.3 | -12.6 | -12.8 | -15.1 | -19.3 | -19.5 | -24.5 | -16.9 |
| | (phase) | (ph5) | (ph6) | (ph6) | (ph6) | (ph5) | (ph5) | (ph5) | (ph5) | (ph5) |
| | -<vdm>[2] | -12.5 | -7.0 | -8.6 | -7.6 | -5.9 | -7.0 | -8.5 | -11.3 | -17.9 |
| | (phase) | (ph7) | (ph7) | (ph7) | (ph7) | (ph7) | (ph7) | (ph6) | (ph6) | (ph5) |

Note: [1] domain (110–130° E, 5–15° S) refer to Table 2 and Fig. 9, and [2] domain (120–
150° E, 0–10° S) refer to Fig. 11.





**Figure List**

**Figure 1.** Wavenumber–frequency spectra for 850-hPa zonal wind averaged over 10° S–10° N in boreal winter after removing the climatological mean seasonal cycle. Vertical dashed lines represent periods at 80 and 30 days, respectively. (a)–(i) are from ERA5 reanalysis, A–CTL, C–CTL, C–1day, C–3days, C–6days, C–12days, C–18days, and C–30days, respectively.

**Figure 2.** Hovmöller diagrams of the correlation between the precipitation averaged over 10° S–5° N, 75–100° E and the intraseasonally filtered precipitation (color) and 850-hPa zonal wind (contour) averaged over 10° N–10° S. (a)–(i) arrange in order are same as Fig. 1 from GPCP/ERA5 and all experiments.

**Figure 3.** Zonal wavenumber–frequency power spectra of anomalous OLR (colors) and phase lag with U850 (vectors) for the symmetric component of tropical waves, with the vertically upward vector representing a phase lag of 0° with phase lag increasing clockwise. Three dispersion straight lines with increasing slopes represent the equatorial Kelvin waves (derived from the shallow water equations) corresponding to three equivalent depths, 12, 25, and 50 m, respectively. (a)–(i) arrange in order are same as Fig. 1 from NOAA/ERA5 and all experiments.

**Figure 4.** Phase-longitude Hovmöller diagrams of 20–100-day filtered precipitation (mm day$^{-1}$, shaded) and SST anomaly (K, contour) averaged over 10° N–10° S from phase 1 to 8. Contour interval is 0.03; solid, dashed, and thick-black lines represent positive, negative, and zero values, respectively. (a)–(i) arrange in order are same as Fig. 1 from GPCP/OISST and all experiments.

**Figure 5.** Phase-vertical Hovmöller diagrams of 20–100-day specific humidity (shading, g kg$^{-1}$) and air temperture (contoured, K) averaged over 10° N–10° S, 120–150° E; solid, dashed, and thick-black curves are positive, negative, and zero values, respectively. (a)–(i) arrange in order are same as Fig. 1 from ERA5 and all experiments.

**Figure 6.** 15° N–15° S averaged p-vertical velocity anomaly (Pa s$^{-1}$, shaded) and zonal wind anomaly (m s$^{-1}$, contour, interval 0.5) between phase 3 and phase 4; solid, dashed, and thick-black lines represent positive, negative, and zero values, respectively.





**Figure 7.** The average 20–100-day filtered oceanic temperature (K, shaded and contour, interval 0.03) between 0 and 60 m depth for MJO phase 2–3. (a)–(g) are from C–CTL, C–1day, C–3days, C–6days, C–12days, C–18days, and C–30days, respectively.

**Figure 8.** The near-equatorial RMM1 and RMM2 variances in a bar graph based on Wheeler and Hendon (2004) with observation and reanalysis data (NOAA/ERA5), AGCM (A–CTL), high-frequency experiments (C–CTL, C–1day and C–3days) and low-frequency experiments (C–6days, C–12days, C–18days and C–30days).

**Figure 9.** The lead-lag relationship between MJO-related atmosphere and sub-seasonal SST variation is examined between phase 1 and 8 within the domain of 110–130° E and 5–15° S. The variables analyzed include 20-100-day filtered latent heat flux (LHF, represented by green shading), outgoing longwave radiation (OLR, represented by yellow bar chart), net surface solar radiation (FSNS, represented by orange bar chart), 850-hPa zonal wind (U850, represented by purple bar chart), 30-m depth oceanic temperature (30-m T multiplied by 100, represented by black line), and sea surface temperatures (SST multiplied by 10, represented by orange line). The graphic expression of variables denoted with (L) indicates the use of the left y-axis, while variables denoted with (R) use the right y-axis. (a)–(i) are from ERA5/OISST reanalysis, A–CTL, C–CTL, C–1day, C–3days, C–6days, C–12days, C–18days, and C–30days, respectively.

**Figure 10.** Phase 4 average 20–100-day filtered OLR (W m$^{-2}$, shaded) and 200 hPa zonal wind anomaly (m s$^{-1}$, vector) with the reference vector (2 m s$^{-1}$) shown at the top right of each panel at the top panel; latent heat flux (W m$^{-2}$, shaded) which positive anomaly represents upward, and 10-m wind anomaly (m s$^{-1}$, contour interval 0.2); solid, dashed, and thick-black lines represent positive, negative, and zero values, respectively, at the second panel from the top, net surface heat flux (W m$^{-2}$, shaded) and net solar radiation (W m$^{-2}$, contour interval 3) at the third panel from the top, and SST (K, shaded) and 850 hPa zonal wind anomaly (m s$^{-1}$, vector) with the reference vector (1 m s$^{-1}$) shown at the top right of each panel at the bottom panel. The number of days used to generate the composite is shown at the bottom right corner of each panel and vertical black line of each panel indicates the dateline. (a), (d), (g) and (j) are from C–CTL; (b), (e), (h) and (k) are from C–18days, and (c), (f), (i) and (l) are from C–30days, respectively.



**Figure 11.** The bar chart illustrates anomalies in the average 20–100-day filtered column-integrated MSE budget terms (J kg$^{-1}$ s$^{-1}$) across the domain (10° S–0° N/S, 120–150° E) for REA5 and all model simulations. Different colors represent different datasets: green for REA5, light gray for A–CTL, red, orange and wathet blue for high-frequency experiments (C–CTL, C–1day, and C–3days), respectively, purple, blue, dark brown, and dark gray for low-frequency experiments (C–6days, C–12days, C–18days, and C–30days), respectively. The bars from left to right represent column-integrated MSE tendency (<dmdt>), column-integrated vertical MSE advection (-<wdmdp>), column-integrated horizontal MSE advection (-<vdm>), surface latent heat fluxes (LH), surface sensible heat fluxes (SH), shortwave radiation fluxes (<SW>), longwave radiation fluxes (<LW>) and residual terms, respectively.

**Figure 12.** Phase 5 anomalies of 20–100-day filtered the column-integrated MSE tendency (J kg$^{-1}$ s$^{-1}$, shading), precipitation (mm d$^{-1}$, contours interval 1.0) and 850-hPa wind (green vector) with the reference vector (2 m s$^{-1}$) based on (a) ERA5, (b) A−CTL, (c) C−CTL, (d) C−1day, (e) C−3days, (f) C−6days, (g) C−12days, (h) C−18days and (i) C−30days. The solid-red, dashed-blue, and thick-black curves represent positive, negative, and zero values, respectively. The vertical black line in each panel indicates the dateline.

**Figure 13.** (a) The relative role of each MSE component of phase 5 through the projection of the spatial pattern of each MSE budget over the MC (20° S–20° N, 90–210° E) onto the total MSE tendency pattern (Fig. 12a). (b–c) Decomposite of the total horizontal MSE advection based on zonal and meridional components of high-frequency SST feedback experiments (C–CTL, A–CTL, C–1day and C–3days) and low-frequency SST feedback experiments (C–6days, C–12days, C–18days and C–30days), respectively.

**Figure 14.** Phase 5 anomalies of 20–100-day filtered the column-integrated vertical MSE advection (J kg$^{-1}$ s$^{-1}$, shading), column-integrated horizontal MSE advection (J kg$^{-1}$ s$^{-1}$, contours interval 6.0) and 200-hPa wind (green vector) with the reference vector (3 m s$^{-1}$) based on (a) ERA5, (b) A−CTL, (c) C−CTL, (d) C−1day, (e) C−3days, (f) C−6days, (g) C−12days, (h) C−18days and (i) C−30days. The solid-blue, dashed-red, and thick-black curves represent positive, negative, and zero values, respectively. The vertical black line in each panel indicates the dateline.

**Figure 15.** The sketch map illustrates the equatorial circulation anomalies and moistening processes during the eastward propagation of the MJO in boreal winter for





various experiments: (a) uncoupled A−CTL, (b) high-frequency SST feedback
experiments (C−CTL, C−1day, and C−3days), (c) low-frequency SST feedback
experiments (C−6days, C−12days, and C−18days), and (d) C−30days experiment. In
each panel, the horizontal line represents the equator. The clustering of gray clouds
(size) indicates the strength of convective organization. A red ellipse indicates
conventionally driven circulation anomalies. In the coupled simulations, light red
(blue) filled ovals represent warm (cold) SST anomalies (SSTA), and a grass green
filled rectangle represents latent heat flux anomalies. Unresolved convective
processes are indicated by black dots representing low-level moisture. Low-level
moisture convergence into the equatorial trough is shown by light blue arrows, while
midlevel moisture advection is represented by left-pointing green arrows. The deeper
colors or thicker lines on the map indicate stronger anomalies of the MJO factors.
Note: The concept of the figure is based on DeMott et al. (2014), as depicted in Fig.
1295   11.



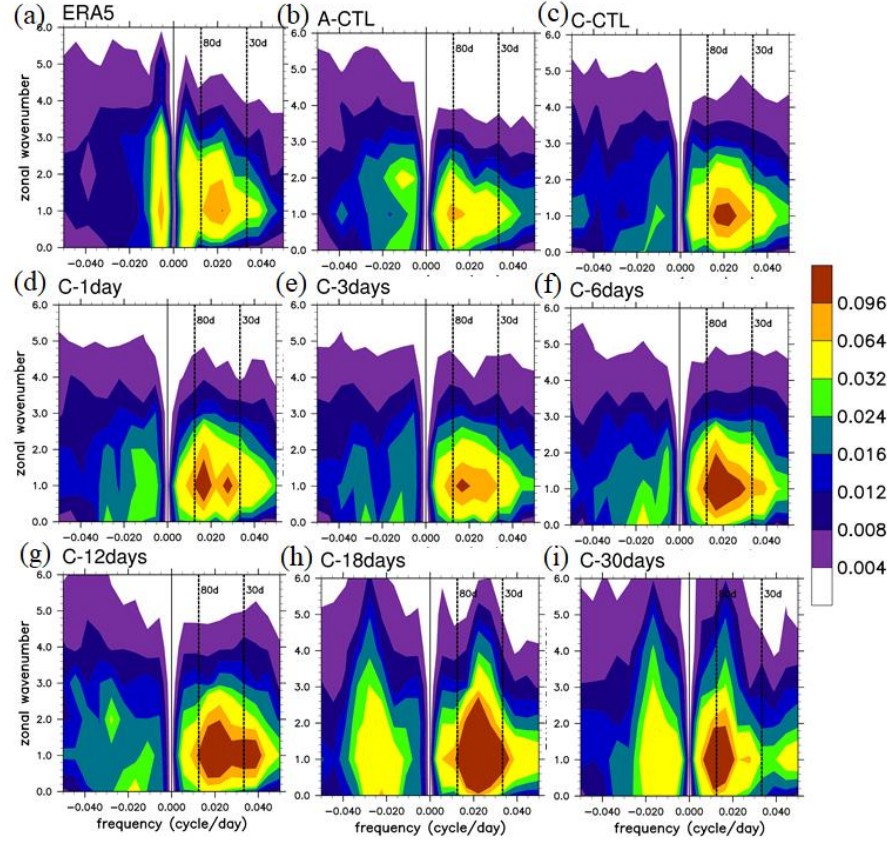

**Figure 1.** Wavenumber–frequency spectra for 850-hPa zonal wind averaged over 10°
S–10° N in boreal winter after removing the climatological mean seasonal cycle.
Vertical dashed lines represent periods at 80 and 30 days, respectively. (a)–(i) are
from ERA5 reanalysis, A–CTL, C–CTL, C–1day, C–3days, C–6days, C–12days, C–
18days, and C–30days, respectively.





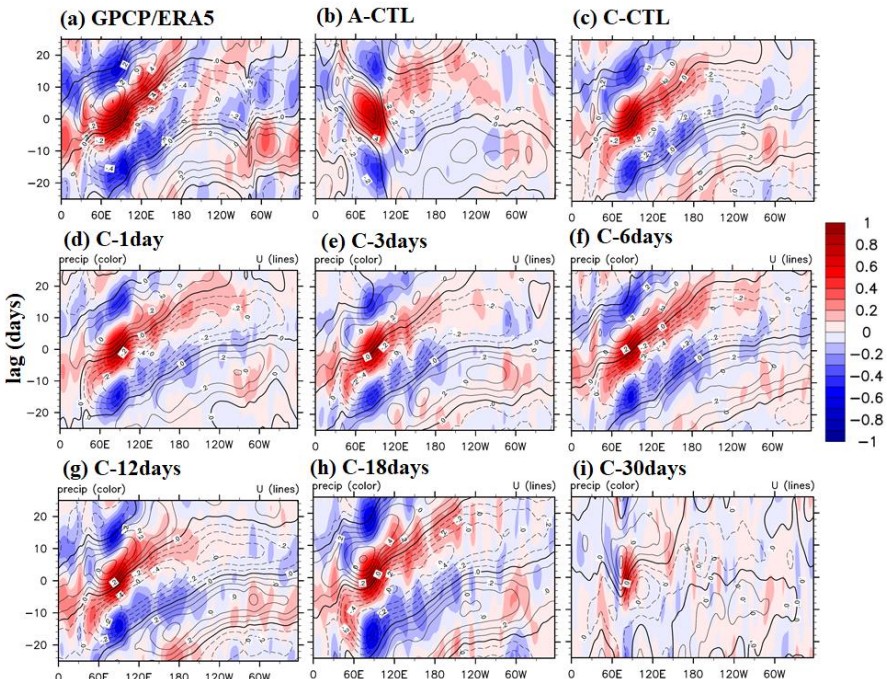

1302

**Figure 2.** Hovmöller diagrams of the correlation between the precipitation averaged over 10° S–5° N, 75–100° E and the intraseasonally filtered precipitation (color) and 850-hPa zonal wind (contour) averaged over 10° N–10° S. (a)–(i) arrange in order are same as Fig. 1 from GPCP/ERA5 and all experiments.

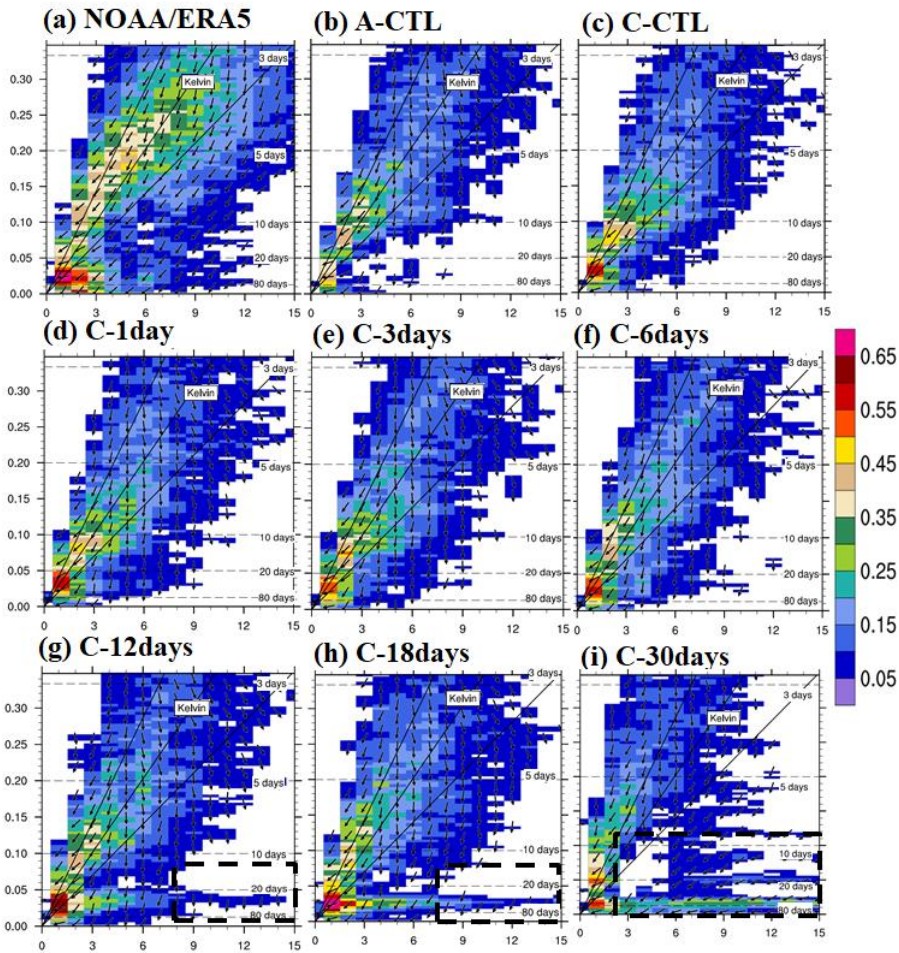

**Figure 3.** Zonal wavenumber–frequency power spectra of anomalous OLR (colors) and phase lag with U850 (vectors) for the symmetric component of tropical waves, with the vertically upward vector representing a phase lag of 0° with phase lag increasing clockwise. Three dispersion straight lines with increasing slopes represent the equatorial Kelvin waves (derived from the shallow water equations) corresponding to three equivalent depths, 12, 25, and 50 m, respectively. (a)–(i) arrange in order are same as Fig. 1 from NOAA/ERA5 and all experiments.



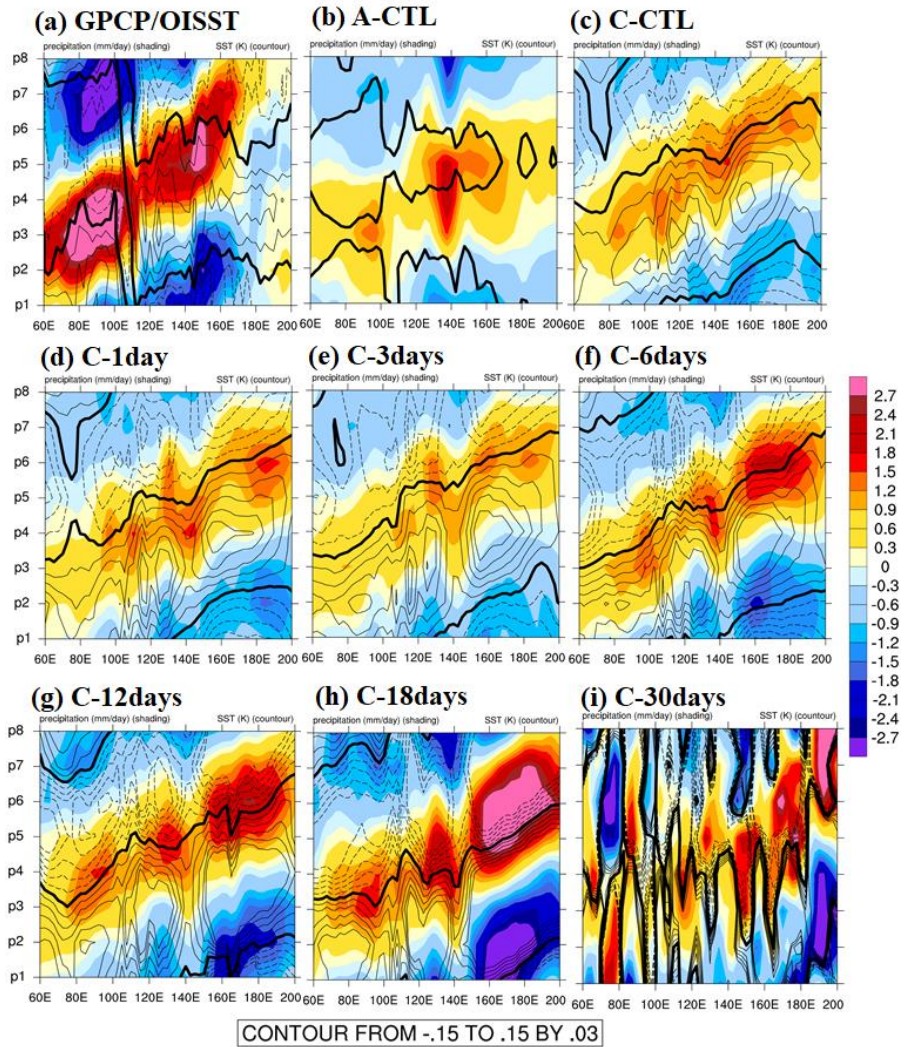

**Figure 4.** Phase-longitude Hovmöller diagrams of 20–100-day filtered precipitation (mm day$^{-1}$, shaded) and SST anomaly (K, contour) averaged over 10° N–10° S from phase 1 to 8. Contour interval is 0.03; solid, dashed, and thick-black lines represent positive, negative, and zero values, respectively. (a)–(i) arrange in order are same as Fig. 1 from GPCP/OISST and all experiments.



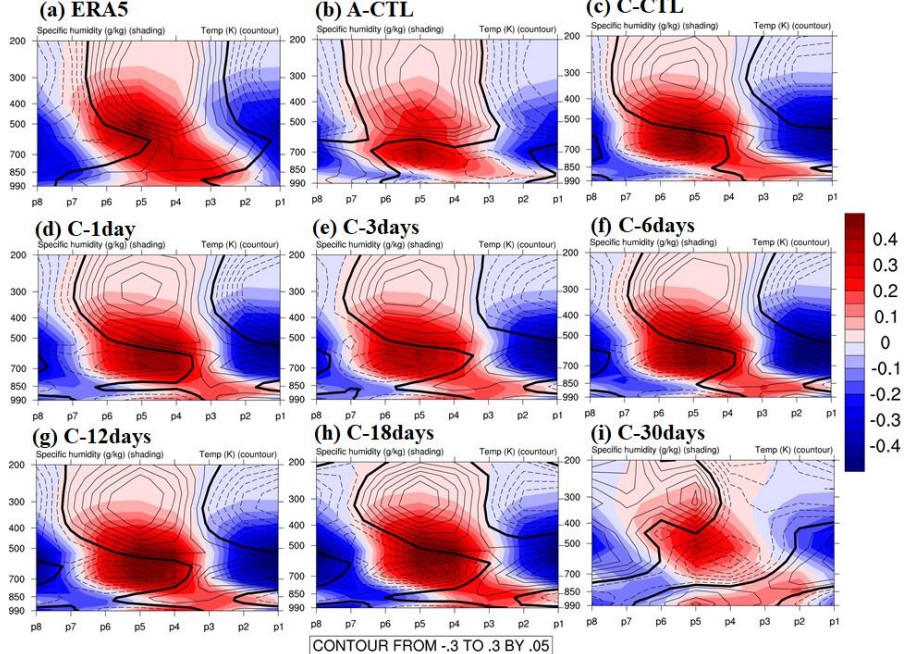

1321

**Figure 5.** Phase-vertical Hovmöller diagrams of 20–100-day specific humidity
(shading, g kg$^{-1}$) and air temperture (contoured, K) averaged over 10° N–10° S, 120–
150° E; solid, dashed, and thick-black curves are positive, negative, and zero values,
respectively. (a)–(i) arrange in order are same as Fig. 1 from ERA5 and all
experiments.



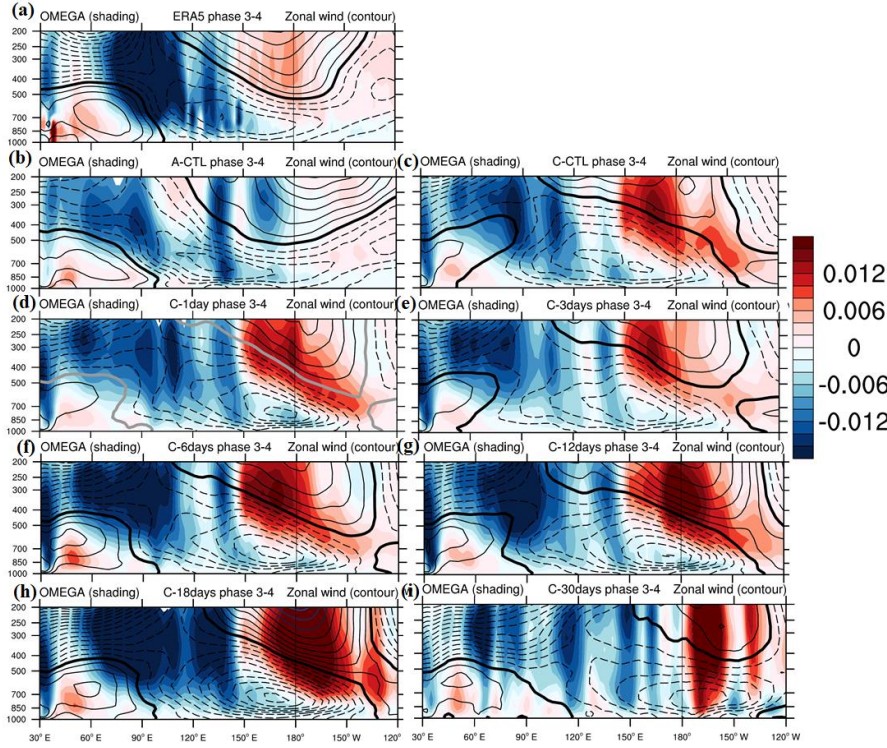

**Figure 6.** 15° N–15° S averaged p-vertical velocity anomaly (Pa s$^{-1}$, shaded) and zonal wind anomaly (m s$^{-1}$, contour, interval 0.5) between phase 3 and phase 4; solid, dashed, and thick-black lines represent positive, negative, and zero values, respectively.




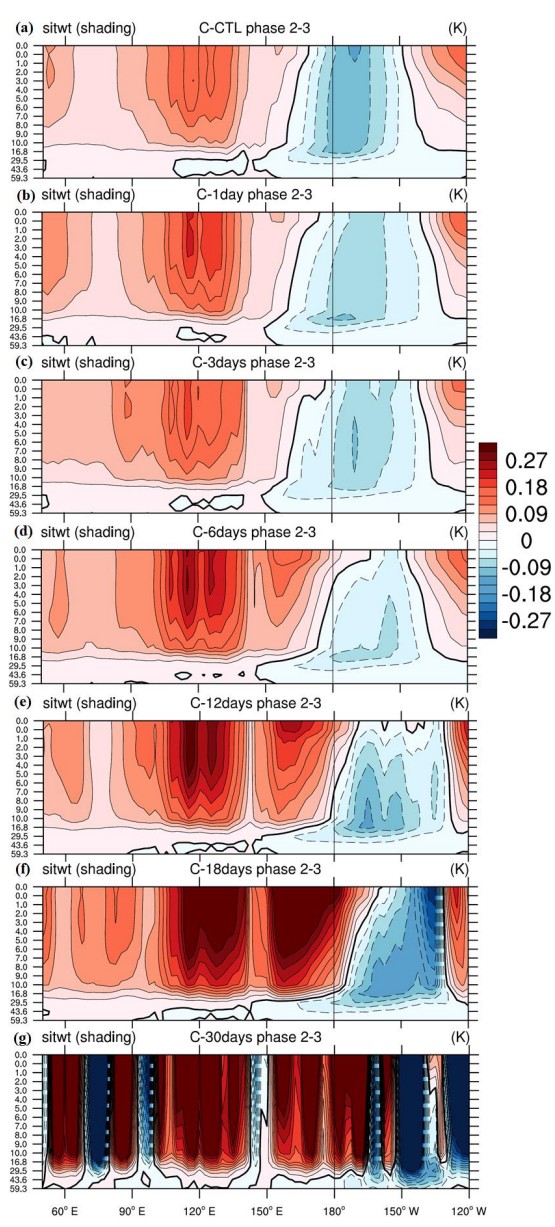

**Figure 7.** The average 20–100-day filtered oceanic temperature (K, shaded and contour, interval 0.03) between 0 and 60 m depth for MJO phase 2–3. (a)–(g) are from C–CTL, C–1day, C–3days, C–6days, C–12days, C–18days, and C–30days, respectively.



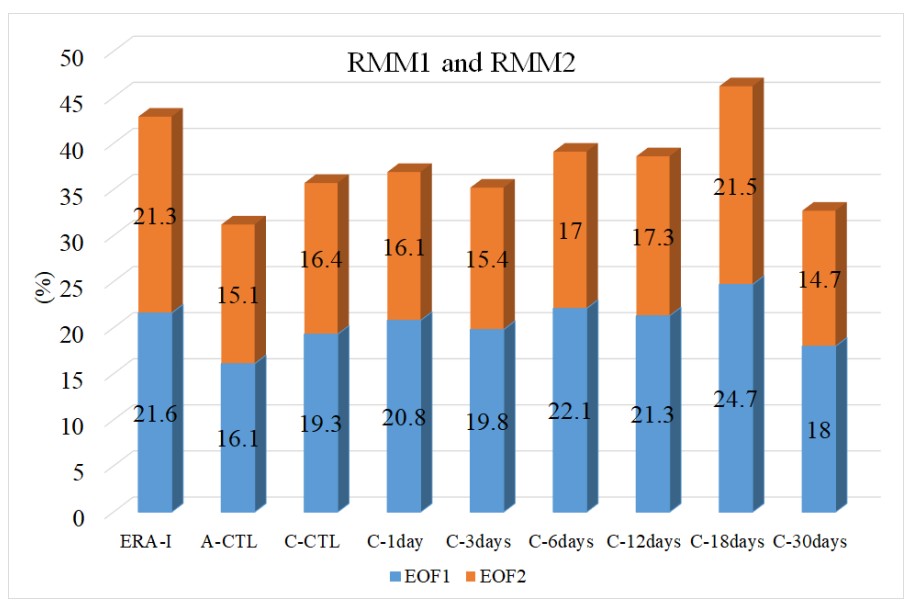

1337

**Figure 8.** The near-equatorial RMM1 and RMM2 variances in a bar graph based on
Wheeler and Hendon (2004) with observation and reanalysis data (NOAA/ERA5),
AGCM (A–CTL), high-frequency experiments (C–CTL, C–1day and C–3days) and
low-frequency experiments (C–6days, C–12days, C–18days and C–30days).

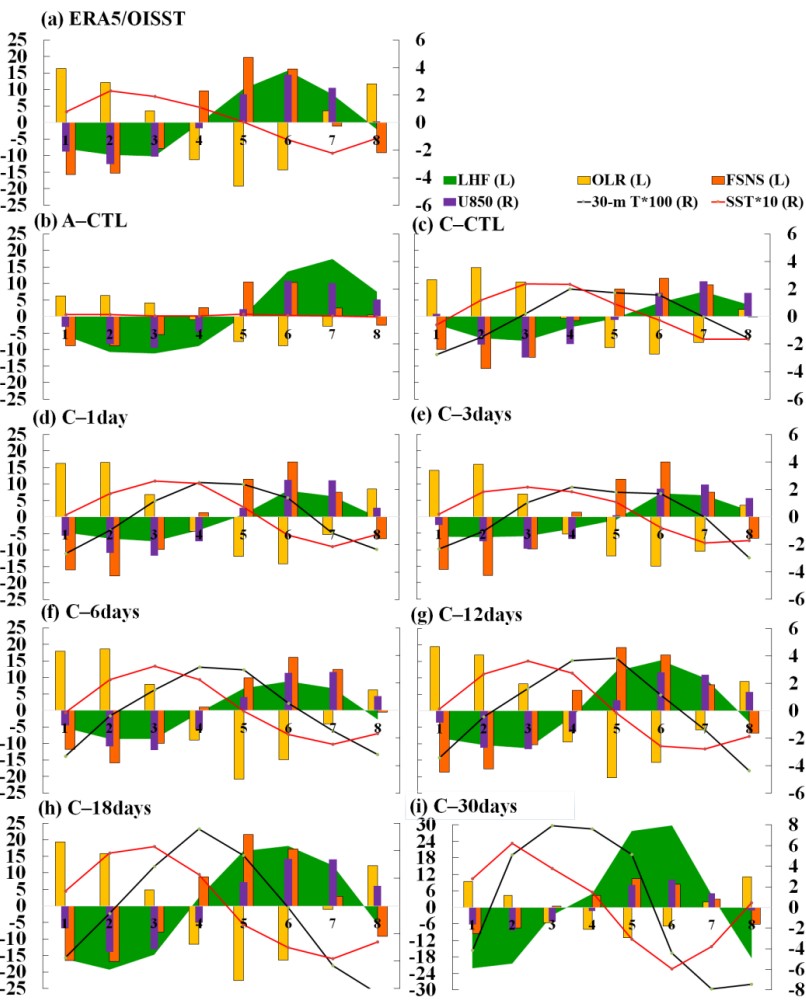

**Figure 9.** The lead-lag relationship between MJO-related atmosphere and sub-seasonal SST variation is examined between phase 1 and 8 within the domain of 110–130° E and 5–15° S. The variables analyzed include 20-100-day filtered latent heat flux (LHF, represented by green shading), outgoing longwave radiation (OLR, represented by yellow bar chart), net surface solar radiation (FSNS, represented by orange bar chart), 850-hPa zonal wind (U850, represented by purple bar chart), 30-m depth oceanic temperature (30-m T multiplied by 100, represented by black line), and sea surface temperatures (SST multiplied by 10, represented by orange line). The graphic expression of variables denoted with (L) indicates the use of the left y-axis, while variables denoted with (R) use the right y-axis. (a)–(i) are from ERA5/OISST reanalysis, A–CTL, C–CTL, C–1day, C–3days, C–6days, C–12days, C–18days, and C–30days, respectively.



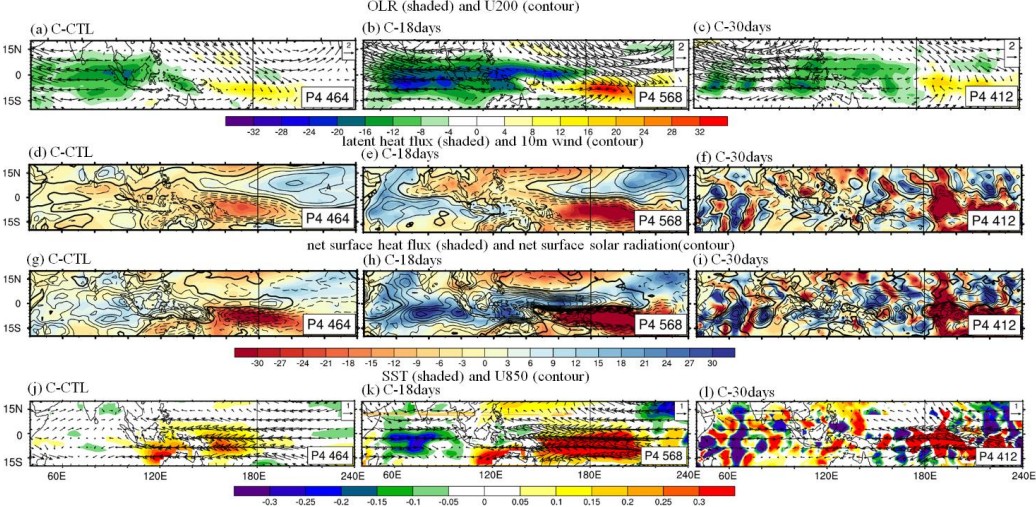

1355

**Figure 10.** Phase 4 average 20–100-day filtered OLR (W m$^{-2}$, shaded) and 200 hPa
zonal wind anomaly (m s$^{-1}$, vector) with the reference vector (2 m s$^{-1}$) shown at the
top right of each panel at the top panel; latent heat flux (W m$^{-2}$, shaded) which
positive anomaly represents upward, and 10-m wind anomaly (m s$^{-1}$, contour interval
0.2); solid, dashed, and thick-black lines represent positive, negative, and zero values,
respectively, at the second panel from the top, net surface heat flux (W m$^{-2}$, shaded)
and net solar radiation (W m$^{-2}$, contour interval 3) at the third panel from the top, and
SST (K, shaded) and 850 hPa zonal wind anomaly (m s$^{-1}$, vector) with the reference
vector (1 m s$^{-1}$) shown at the top right of each panel at the bottom panel. The number
of days used to generate the composite is shown at the bottom right corner of each
panel and vertical black line of each panel indicates the dateline. (a), (d), (g) and (j)
are from C–CTL; (b), (e), (h) and (k) are from C–18days, and (c), (f), (i) and (l) are
from C–30days, respectively.

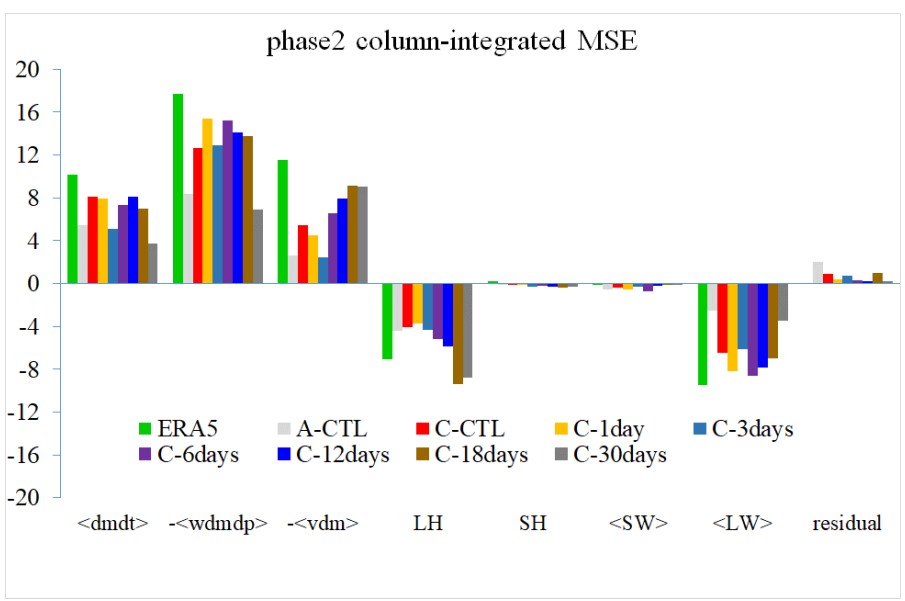

**Figure 11.** The bar chart illustrates anomalies in the average 20–100-day filtered column-integrated MSE budget terms (J kg$^{-1}$ s$^{-1}$) across the domain (10° S–0° N/S, 120–150° E) for REA5 and all model simulations. Different colors represent different datasets: green for REA5, light gray for A–CTL, red, orange and wathet blue for high-frequency experiments (C–CTL, C–1day, and C–3days), respectively, purple, blue, dark brown, and dark gray for low-frequency experiments (C–6days, C–12days, C–18days, and C–30days), respectively. The bars from left to right represent column-integrated MSE tendency (<dmdt>), column-integrated vertical MSE advection (-<wdmdp>), column-integrated horizontal MSE advection (-<vdm>), surface latent heat fluxes (LH), surface sensible heat fluxes (SH), shortwave radiation fluxes (<SW>), longwave radiation fluxes (<LW>) and residual terms, respectively.



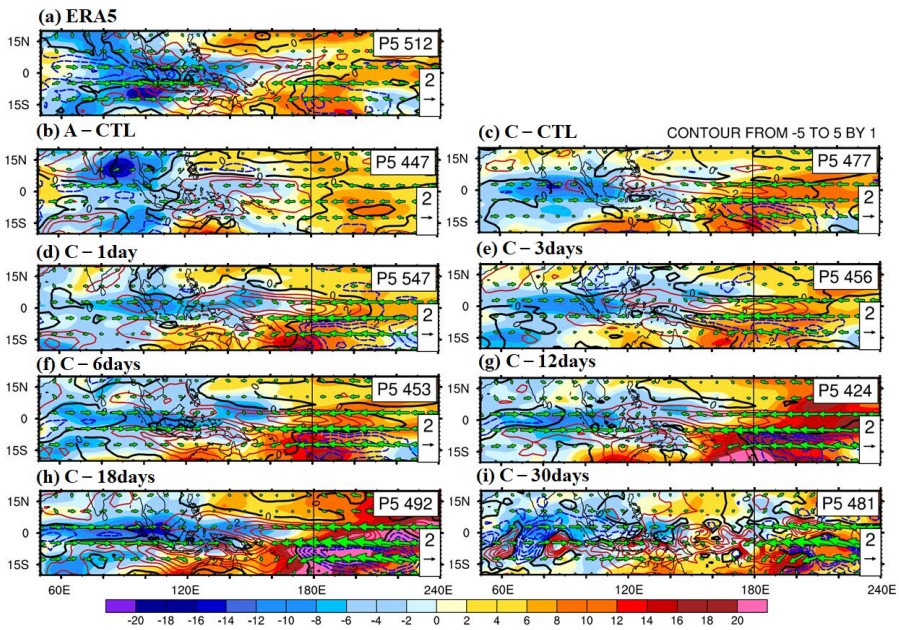

**Figure 12.** Phase 5 anomalies of 20–100-day filtered the column-integrated MSE tendency (J kg$^{-1}$ s$^{-1}$, shading), precipitation (mm d$^{-1}$, contours interval 1.0) and 850-hPa wind (green vector) with the reference vector (2 m s$^{-1}$) based on (a) ERA5, (b) A−CTL, (c) C−CTL, (d) C−1day, (e) C−3days, (f) C−6days, (g) C−12days, (h) C−18days and (i) C−30days. The solid-red, dashed-blue, and thick-black curves represent positive, negative, and zero values, respectively. The vertical black line in each panel indicates the dateline.



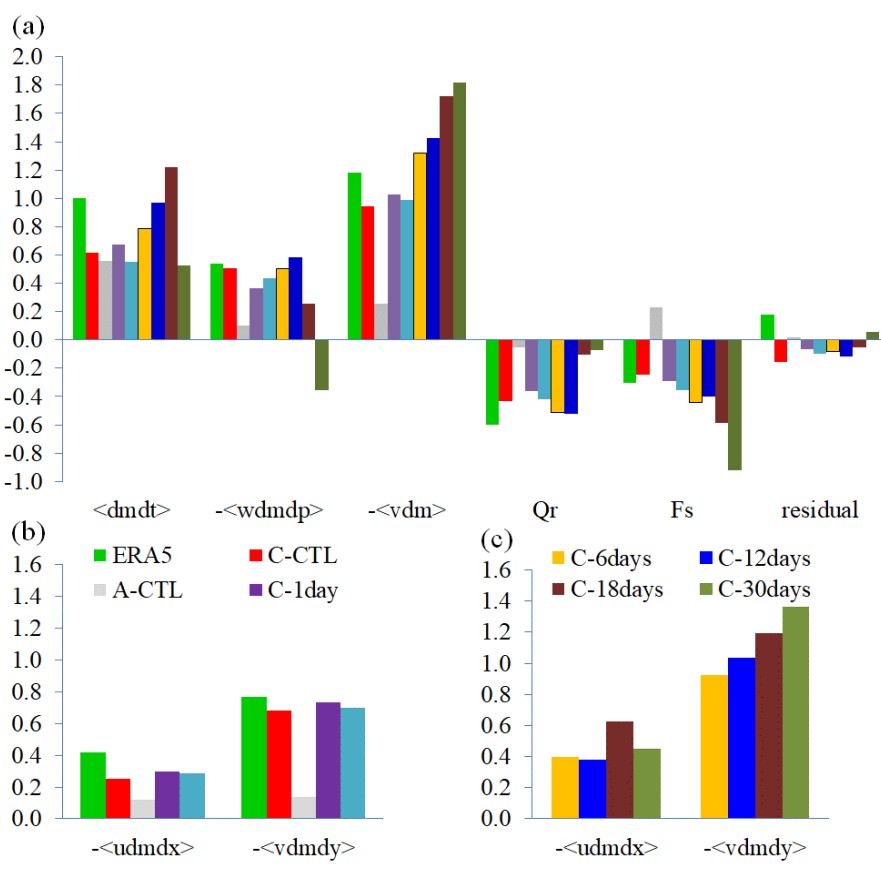

**Figure 13.** (a) The relative role of each MSE component of phase 5 through the projection of the spatial pattern of each MSE budget over the MC (20° S–20° N, 90–210° E) onto the total MSE tendency pattern (Fig. 12a). (b–c) Decomposite of the total horizontal MSE advection based on zonal and meridional components of high-frequency SST feedback experiments (C–CTL, A–CTL, C–1day and C–3days) and low-frequency SST feedback experiments (C–6days, C–12days, C–18days and C–30days), respectively.



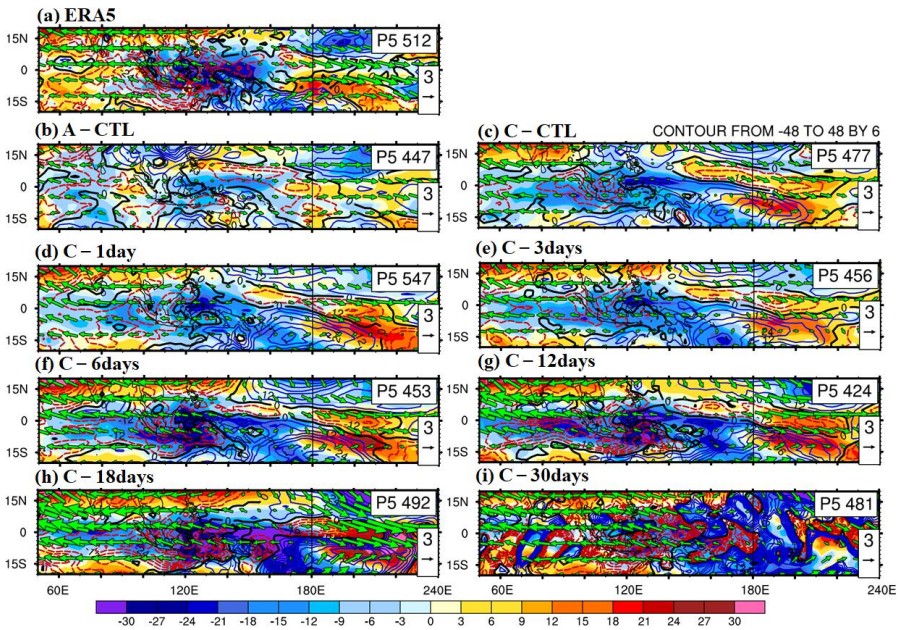

**Figure 14.** Phase 5 anomalies of 20–100-day filtered the column-integrated vertical MSE advection (J kg$^{-1}$ s$^{-1}$, shading), column-integrated horizontal MSE advection (J kg$^{-1}$ s$^{-1}$, contours interval 6.0) and 200-hPa wind (green vector) with the reference vector (3 m s$^{-1}$) based on (a) ERA5, (b) A−CTL, (c) C−CTL, (d) C−1day, (e) C−3days, (f) C−6days, (g) C−12days, (h) C−18days and (i) C−30days. The solid-blue, dashed-red, and thick-black curves represent positive, negative, and zero values, respectively. The vertical black line in each panel indicates the dateline.





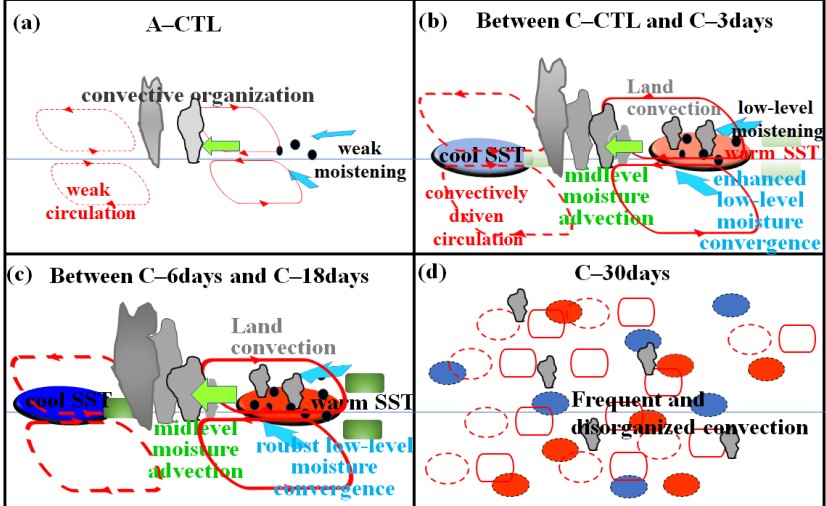

1405

**Figure 15.** The sketch map illustrates the equatorial circulation anomalies and
moistening processes during the eastward propagation of the MJO in boreal winter for
various experiments: (a) uncoupled A−CTL, (b) high-frequency SST feedback
experiments (C−CTL, C−1day, and C−3days), (c) low-frequency SST feedback
experiments (C−6days, C−12days, and C−18days), and (d) C−30days experiment. In
each panel, the horizontal line represents the equator. The clustering of gray clouds
(size) indicates the strength of convective organization. A red ellipse indicates
conventionally driven circulation anomalies. In the coupled simulations, light red
(blue) filled ovals represent warm (cold) SST anomalies, and a grass green filled
rectangle represents latent heat flux anomalies. Unresolved convective processes are
indicated by black dots representing low-level moisture. Low-level moisture
convergence into the equatorial trough is shown by light blue arrows, while midlevel
moisture advection is represented by left-pointing green arrows. The deeper colors or
thicker lines on the map indicate stronger anomalies of the MJO factors. Note: The
concept of the figure is based on DeMott et al. (2014), as depicted in Fig. 11.