# Peer review of "To quantify the impact of SST feedback periodicity on # 2 atmospheric intraseasonal variability in the tropical regions"

_Geoscientific Model Development, 2023_

## Author Comment (AC1)

We greatly appreciate reviewer's insightful and helpful comments regarding our manuscript. The manuscript has been revised based on reviewer's comments. In general, the revised manuscript has been reduced by 15 pages, including the excessive detail in Table 3 and two figures of experimental sensitivity. The missing x- and y-axes labels and cluttered figures have been improved to enhance readability. In addition, we added another experiment with SST feedback at the 24th day (1/24days), which illustrates the transition from unrealistically overestimated MJO with an 18days feedback to poorly-organized MJO with a 30days feedback. Below are the point-by-point replies to reviewer's comments and concerns.

Sincerely,
Yung-Yao Lan, Huang-Hsiung Hsu and Wan-Ling Tseng

Anonymous Referee #1
The reviewer comments are formatted in italics and the authors response to the comments are formatted in bold.
Notation *RC1.P#* represents Reviewers Comment. Paragraph Number

**General major/minor comments:**
* * *
*RC1. I felt that the length of the paper can be reduced to make it more concise. Discussions in many places are redundant, and not necessary. For example, descriptions of the SST influences on the MJO in the lines of 249-256 can be integrated with the introduction part, not necessarily to mention this again in section 3.1; also Lines 490-492 on the leading EOF modes. Similarly for many other places as also further mentioned below.*
* * *
**Response:**

**Thank you for your suggestion. We reduced the length of all sections and made**

**the manuscript more concise.**
* * *
*RC2. I also had a general question on the determination of MJO phases based on the WH04 approach for different model simulations. Since the combined EOF for OLR, u850, u200 is conducted separately for different model experiments, how to make sure all the leading combined EOF modes from the experiments are same with the observations. Otherwise, this will lead to phase differences among model simulations and observations.*
* * *
**Response:**

**Wheeler et al. (2004) projected daily OLR, u850, and u200 onto the multiple-**

**variable EOFs, with the annual cycle and components of interannual variability**

**removed, results in principal component (PC) time series under the same criteria**

**that primarily vary on the intraseasonal time scale of the MJO. If the leading**

**EOF modes from the model experiments align well with the observations, it**

**suggests that the model is skillfully representing the MJO's behavior. We**

**followed the same procedures and applied to the observations and all model**

simulations. In general, the spatio-temporal atmospheric structures in model experiments closely resemble those observed, although minor differences might be observed. The consistency is evident in the Hovmöller diagrams showing the evolution of the MJO. Reviewer's concern is well taken but it does not seem to be a problem in our study.

Wheeler, M. C., and Hendon, H. H.: An all-season real-time multivariate MJO index: development of an index for monitoring and prediction, Mon. Weather Rev., 132, 1917–1932, https://doi.org/10.1175/1520-0493(2004)132<1917:AARMMI>2.0.CO;2, 2004.

> RC3. One of the main findings from this study is that "The increasing feedback periodicity of SST in low-frequency experiments leads to the accumulation of short-wave and long-wave radiations and surface heat fluxes from the atmosphere, resulting in an increase in the upper oceanic temperature and its variances (Lines 813-816)". Although this seems supported by the results in this study, I do not completely understand why this is the case. My understanding is that for an intraseasonal time scale, such as 30 days corresponding to the coupling time-scale in the C-30days experiment, the variations in radiation or heat fluxes can be in positive and negative phases, so they can be cancelled out if averaged over 30 days - not necessary always accumulating large positive or negative values.

**Response:**

Thank you for this valuable question. To demonstrate "SST feedback periodicity impact on MJO," we designed asymmetric exchange frequencies between the atmosphere and the ocean as outlined in Table 1. In these experiments, the ocean continuously receives atmospheric forcing at every time step, but the ocean's feedback to the atmosphere is only allowed at certain time intervals, including 30 minutes, 1 day, 3 days, 6 days, 12 days, 18 days, 24 days, and 30 days. The C–30days experiment can be conceptualized as the atmosphere continuously forcing the ocean (through radiation and heat fluxes), while the ocean does not respond to the atmosphere until 30 days have passed.

As shown in Figure 2, 4, and 7, lower feedback frequency experiments (especially C-24days and C-30days) tended to simulate more stationary (weaker propagating tendency) MJO. That means the atmospheric forcing tended to be more in the same sign for a longer period. This feedback could continue to accumulate or

deplete the heat in the model ocean, especially in the subsurface. When the coupling frequency reduced to lower than that of the MJO fluctuations, such as 1/30days, the atmosphere-ocean coupling in the MJO failed to work and the results became unrealistic. To further demonstrate  this, we conducted an additional experiment with SST feedback at the 24th day (1/24days), which yielded a result transitioning from C-18days to C-30days (refer to Table 2, and Fig. 1−12).

We agree with the reviewer that *"the variations in radiation or heat fluxes can be in positive and negative phases"* and this is likely why the ocean temperature and heat flux distributions break into small-scale unorganized structure. We added plots for phase 4−5 in revised Fig. 6, which exhibits more negative ocean temperature anomalies in the Indian Ocean under prevailing westerly anomalies, in contrast to more positive anomalies in phase 2-3. Positive and negative radiation and heat fluxes are likely not completely cancelled out as suspected by the reviewer, resulting in small scale horizontal distribution. This discussion is added in the revised manuscript.

> *RC4. Several figures look very busy, such as Figs. 12, 14, could be further improved to make them more readable.*

**Response:**

In revised manuscript of Figs. 8, 10, and 12, we have reduced the color contrast, changed the whiteness near 0 in shading, increased the contour interval, and reduced vector density to create a less cluttered appearance. Additionally, in Fig. 13, we have simplified the information presented.

**Other comments:**

> *RC5. Line 9: what is "TKE"?*

**Response:**

Thank you for your comment. The acronym "TKE" stands for turbulence kinetic energy. The additional sentences have been integrated into the revised manuscript.

> *RC6. Line 27: what is "lower results"? You meant weaker amplitude?*

**Response:**

**Yes, this sentence indicates that the slightly underestimated compared to ERA5 and NOAA data.**

> *RC7. Line 90: what is "dynamic range"?*

**Response:**

**Thank you for your comment. We mean a reduced intraseasonal SST variability.**

> *RC8. Line 99: "weakness"?*

**Response:**

**We have changed "weakness" to "weak".**

> *RC9. Line 100: "Understanding the manifestation of heat fluxes in ...". I don't quite follow this sentence.*

**Response:**

**Analysis of the intraseasonal oscillation (ISO) reveals that heat fluxes play a critical role in the development of intraseasonal SST variability (Liang et al., 2018). Please see lines 34−36.**

> *RC10. Line 119: Jiang (2017), Gonzalez and Jiang (2017) are very relevant to the discussions here on the relationship between the mean MSE/moisture pattern and MJO propagation.*
>
> *Jiang, X., 2017: Key processes for the eastward propagation of the Madden-Julian Oscillation based on multimodel simulations. Journal of Geophysical Research: Atmospheres, 10.1002/2016JD025955.*
>
> *Gonzalez, A. O. and X. Jiang, 2017: Winter Mean Lower-Tropospheric Moisture over the Maritime Continent as a Climate Model Diagnostic Metric for the Propagation of the Madden-Julian Oscillation. Geophys. Res. Lett., 10.1002/2016GL072430.*

**Response:**

**Thank you for the suggestion. Their references have been included to enhance this manuscript's introduction and discussion of the relationship between the mean MSE/moisture pattern and MJO propagation. Please see lines 602−607.**

**Response:**

**Thank you for your comment. Please see the revised manuscript for the change.**

> *RC12. Line 122-123: On the first order, the PBL convergence ahead of the MJO convection is due to Kelvin-wave dynamics, rather than SST induced.*

**Response:**

**Thank you for your comment. Please see lines 602−605 for the change.**

> *RC13. Line 127: what is the "positive trend" being discussed here? This sentence needs to be improved.*

**Response:**

**A 'positive trend' refers to changes in vertical MSE advection are likely responsible for the increase in MJO variability with SST.**

> *RC14. Line 188: is there a gradual transition belt between the coupled and uncoupled zones?*

**Response:**

**No, the presence of a gradual transition belt defined between 30-40° N and 30-40° S tended to weaken the simulated MJO. None of the experiments in this study include a gradual transition belt between the coupled and uncoupled zones. The setting closely resembles that of Lan et al. (2022) and Tseng et al. (2022).**

**Lan, Y.-Y., Hsu, H.-H., Tseng, W.-L., and Jiang, L.-C.: Embedding a one-column ocean model in the Community Atmosphere Model 5.3 to improve Madden−Julian Oscillation simulation in boreal winter, Geosci. Model Dev., 15, 5689−5712, https://doi.org/10.5194/gmd-15-5689-2022, 2022.**

**Tseng, W.-L., Hsu, H.-H., Lan, Y.-Y., Lee, W.-L., Tu, C.-Y., Kuo, P.-H., Tsuang, B.-J., and Liang, H.-C.: Improving Madden−Julian oscillation simulation in atmospheric general circulation models by coupling with a one-dimensional snow−ice−thermocline ocean model, Geosci. Model Dev., 15, 5529−5546, https://doi.org/10.5194/gmd-15-5529-2022, 2022.**

> *RC15. Lines 204-205: is there a reason for the different nudging time-scale for different depth in the ocean?*

**Response:**

**A strong nudging in the deeper region was designed to prevent the model ocean temperature drift because of lack of ocean circulation. By contrast, the much weaker nudging in the upper 10.5 m to 107.8 m allows the ocean to respond efficiently to the surface fluctuations, which were coupled without nudging to allow fast response to the atmospheric forcing and enhance the atmosphere-ocean coupling.**

> *RC16. Line 249: "interseasonal" needs to be fixed here, as well as in several other places in the paper*

**Response:**

**Thank you for pointing out the errors that have been corrected.**

> *RC17. Line 251: "the behavior of the MJO behavior"….*

**Response:**

**Thank you for pointing out the error. This sentence was integrated with the introduction part to make it more concise.**

> *RC18. Lines 256-257: How the "Cooler than average SST to the east of MJO convection is associated with the passage of the MJO"?*

**Response:**

**It has been corrected. Thank you for the suggestion.**

> *RC19. Lines 286-287: "the atmospheric heat/cooling"?*

**Response:**

**The maximum (minimum) $T_{10m}$ values lagging 1 phase behind SST indicate that the atmospheric heating (cooling) ocean process in the high-frequency experiments, but not in the low-frequency experiments.**

> *RC21. Fig. 3: Labels for x- and y-axes are missing*

**Response:**

**The x- and y-axes are added in Figs. 3, 4, 5, 6, 7, 9, and 11 in revised manuscript.**

> *RC22. Line 366: suggest change "MJO" to "intraseasonal", since the MJO band is within wavenumber 1-5.*

**Response:**

**Thank you for your comment. Please see line 266 for the change.**

> *RC23. Line 367-368: this sentence needs to be connected with the following sentences.*

**Response:**

**We combine the two sentences as follows: Figure 4 show the phase–longitude diagrams in which the 20–100 d filtered precipitation (shaded) and SST (contour) anomalies were averaged over the region from 10° S to 10° N to determine the relationship between precipitation and SST fluctuations and to provide insights into the connection between air–sea coupling and convection.**

> *RC24. Line 383: suggest change "became" to "becomes"*

**Response:**

**Thank you for your suggestion. Please see line 233 for the change.**

> *RC25. Lines 403-405: in ERA5, why the warm T near surface is located to the west of MJO, seems not consistent with the SST.*

**Response:**

**Thank you for your feedback. In original Fig. 5, we present phase-vertical Hovmöller diagrams illustrating specific humidity (shading, in g kg$^{-1}$) and air temperature (contoured, in K) averaged over a fixed 10° N–10° S, 120–150° E region for the 20–100-day period. The x-axis denotes phases from 1 to 8, arranged from right to left, does not represent area located to the west or east of the MJO. Kim et al. (2017) indicated eastward-propagating MJO detours southward in the MC region, exhibiting enhanced convective activity preferentially in the southern MC area, with weaker anomalies in the central MC area. During phases 6–7 (Fig. RC#1.1), the air temperature at a 2m height exhibits a positive anomaly in the northern part of this area but a negative**

To avoid this weak surface temperature anomalies in the central MC area, we have reselected data points aligning with enhanced convective activity in the southern MC area (5–20° S, 120–150° E), ensuring consistency between surface temperature and surface heat flux as shown in the revised Fig. 5. Figure RC#1.1 illustrates a comparison between the central MC area and the southern MC area in phase-vertical Hovmöller diagrams, highlighting specific humidity (shading, in g kg⁻¹) and air temperature (contoured, in K).

[Figure]

Figure RC#1.1 A comparison between averaged phase-vertical Hovmöller diagrams of 20–100-day specific humidity (shading, g kg⁻¹) and air temperature (contoured, K) for two latitudinal bands: (1) 10° N–10° S and (2) 5–20° S over 120–150° E area; solid, dashed, and thick-black curves represent positive, negative, and zero values, respectively.

> *RC26. Lines 423-425: Does this sentence mean the land convection over the MC is critical for MJO eastward propagation over the MC region? But in the reality, when MJO propagates over MC, the active MJO convection is largely over the oceanic region, while the convection over the land is suppressed. If you want to emphasize this point, may need to provide more evidence. Similar statements were also discussed in the conclusion part.*

**Response:**

**Thank you for your comment. Most of research journals, such as those by Ahn et al. (2020), Savarin and Chen (2023), and Zhang and Han (2020), have discussed the Maritime Continent barrier effect on MJO propagation. We initially observed that land convection over the MC leads the major convection, acting like a precursor, although we lacked further evidence to support this claim. In the revised manuscript, we have removed the discussion of land convection to avoid introducing confusion and to maintain focus on the primary subject of this**

**study.**

Ahn, M., Kim, D., Ham, Y., and Park, S.: Role of Maritime Continent land convection on the mean state and MJO propagation, J. Clim. 33:1659–1675, https://doi.org/10.1175/JCLI-D-19-0342.1, 2020.

Savarin, A., and Chen, S. S.: Land-locked convection as a barrier to MJO propagation across the Maritime Continent, J. Adv. Model. Earth Syst., 15, e2022MS003503. ttps://doi.org/10.1029/2022MS003503, 2023.

Zhang, L., and Han, W.: Barrier for the eastward propagation of Madden-Julian Oscillation over the Maritime Continent: A possible new mechanism. Geophys. Res. Lett., 47, e2020GL090211. https://doi.org/10.1029/2020GL090211, 2020.
* * *
RC27. Fig. 6d The grey color for the thick contours needs to be corrected for consistency.

**Response:**

**It has been corrected. Thank you for the suggestion.**
* * *
RC28. Lines 451-463: discussions in this part are largely reductant with previous discussions, particularly in the introduction part, so can be significantly reduced to be concise.

**Response:**

**Thank you for the suggestion. A concise discussion of this aspect can be found in section 3.2.3.**
* * *
RC29. Line 465: is vertical or horizontal gradient discussed here?

**Response:**

**The 1-D TKE ocean model only considers the vertical gradient of temperature in the upper ocean.**
* * *
RC30. Line 475: "a cooling effect in the upper ocean" mentioned here is not obvious to me

**Response:**

**When the MJO convection moves across the IO (60–90° E), the westerlies extracted heat from the ocean surface, resulting in cooling tendency (i.e., 0.1 K from phase 1 to 3 and further cooling in phase 5 as shown in Fig. RC#1.2) of the**

**IO near surface temperature in the C–CTL.**

[Figure]

**Figure RC#1.2 The evolution of filtered oceanic temperature anomalies (K) for C–CTL, averaged over the depth of 0–60 meters, within the 0°–15° S latitude range at phases 1, 3, 5, and 7.**

> *RC31. Line 477: this sentence can be improved – "…characterized by stronger intraseasonal MJO variability" is for C-CTL or C-3days?*

**Response:**

**Stronger intraseasonal MJO variability is for C-CTL. Please see the revised manuscript for the change.**

> *RC32. Lines 488-500: discussions here can also be more concise since these have been discussed earlier.*

**Response:**

**The EOF analysis discussion is removed. A more concise discussion can be found in the revised manuscript.**

> *RC33. Lines 556-559: Just curious that the phase delay of 30-m T relative to surface T seems not seen in Fig. 7. Any thoughts on this?*

**Response:**

**The revised Fig. 6 displays the spatial distribution of oceanic temperature between 0 and 60 m depth for the average of phases 2 and 3 in all experiments. Time evolution is not considered in revised Fig. 6, making it unable to depict the phase delay between the 30-meter temperature and SST. The 30-m temperature anomaly exhibits a one-phase delay compared to SST, indicating that MJO convection extracts heat from the ocean surface, and vertical mixing requires time to propagate downward. This delay effect is also evident in the field campaign; de Szoeke et al. (2015) observed that the ocean warmest 10-m temperature occurred a few days later than the peak temperature at 0.1 m. Additionally, the 0.1-m ocean temperature was typically as warm as or warmer than the 10-m temperature.**

**de Szoeke, S. P., Edson, J. B., Marion, J. R., Fairall, C. W., and Bariteau, L.: The MJO and air–sea interaction in TOGA COARE and DYNAMO, J. Climate, 28, 597–622, https://doi.org/10.1175/JCLI-D-14-00477.1, 2015.**

> *RC34. Lines 632-637: This part again can be more concise since these have been previously mentioned. In general, I felt that the Section 4.4 is a bit lengthy, can be more concise and make important points more clearly delivered.*

**Response:**

**Thank you for your suggestion. We reduced the length of all sections and made it more concise in the revised manuscript.**

> *RC35. Fig. 13b,c: why not combine these two panels together as in the upper panel?*

**Response:**

**Thank you for the suggestion. We showed it in revised Fig. 11.**

[Figure]

**Figure 11. The projection of each MSE component (<dmdt>, -<wdmdp>, -<vdm>, Qr, Fs and residual) and decomposition of the total horizontal MSE advection (- and -<vdmdy>) at phase 5 over the MC (20° S–20° N, 90–210° E) onto the ERA5 column-integrated MSE tendency (Fig. 10a).**

*RC36. Line 778: suggest change "ERA5" to "observations" since this also involves GPCP and SST data.*

**Response:**

**Thank you for your suggestion. Please see the revised manuscript for the change.**

*RC37. Lines 773-786: In this part, when mentioning the related figures, may just provide the figure number, for example, just use "(Fig. 1)" instead of "(as shown in Fig. 1)" in Line 778, and similarly for many others.*

**Response:**

**Thank you for your comment. Please see lines 542–547 for the change.**

*RC38. Line 790: suggest change "oceanic heat fluxes" to "surface heat fluxes"*

**Response:**

**Thank you for your suggestion. Please see line 556 for the change.**

*RC39. Lines 802-805: "… it becomes evident that the high frequency (low-frequency) SST experiments tended to underestimate (overestimate) the MJO simulation ". Just wonder if this statement can be model-dependent?*

**Response:**

**The underestimation in the high frequency SST experiments might be a deficit in the model, which was gradually recovered and become overestimated due likely to the increasing heat accumulation in the ocean with decreasing SST frequency as discussed in previous parts of the manuscript.**

---

## Author Comment (AC2)

We greatly appreciate reviewer's insightful and helpful comments on our manuscript. The manuscript has been revised based on reviewer's comments. The revised manuscript has been reduced by 15 pages, by condensing the description of experimental sensitivity and repeated discussion. The revised manuscript is now more concise than the previous version.

Sincerely,
Yung-Yao Lan, Huang-Hsiung Hsu and Wan-Ling Tseng

Anonymous Referee #2
The reviewer comments are formatted in italics and the authors response to the comments are formatted in bold.
Notation *RC2.P#* represents Reviewers Comment. Paragraph Number

**Major comments:**

*RC1. My main comment is about the experimental setup. The "frequency" that is varied here is the frequency of the update of the SST seen by the atmospheric model. While the SST is held fixed, the ocean-model SST evolves in time in response to the surface heat fluxes and vertical mixing. When the SST is updated for the atmospheric model, it increases or decreases abruptly. The less frequent the update, the larger the potential jump in SST. This is probably why, as the frequency is decreased down to 1/(30 days), there is a lot of unorganized variability rather than a simulation that becomes similar to the atmosphere-only simulation, as one could have expected if this experimental setup actually filtered the subseasonal variability of SST (I expect the atmosphere-only experiment to use the classical configuration with smoothly varying SST). The SST jumps seen by the atmosphere raise a number of questions. Potentially, they could themselves, occasionally, trigger the development of an intraseasonal convective disturbance, in which case they become a forcing instead of expressing a feedback. Otherwise, they happen at fixed dates and one can wonder how they can be properly phased with an intraseasonal disturbance. The case with an SST-updating timestep of 18 days is probably optimum because it corresponds to half the period of the MJO: locally, a warm SST jump ushers the development phase of the MJO which extracts energy from the surface, and the downward SST jump starts its decline. But does the ocean feedback on the atmosphere in this case, as we expect for the MJO, or does the atmosphere feedback on the SST jumps? The fact that the signal is so strong is suspicious: considering because there is also a sampling problem associated with this experimental setup: with an SST-update timestep of 18 days and the simulation duration of 30 years, the SST update happens less than twice at the same date, and there are rarely more than 4 MJO events per year. To get a very clear signal, we need most of the SST updates to occur within a couple of days of the maximum or minimum convective activity. This seems unlikely to me considering the small numbers of update days at a given date and the expected number of spontaneously-generated MJO events. The probability that the SST jumps time the MJO events is non-negligible, which probably means that the SST plays some role in forcing these events, instead of providing a feedback to these events.*

*For this reason, there is a need to better understand the impact of the experimental setup. This could be done by looking at singular MJO events rather than composites and/or conducting additional experiments with different configurations (updating the SST seen by the atmospheric model with smoothed tendencies using the tendency history, maybe).*

**Response:**

Thank you for your comments. Reviewer's comments are insightful and well taken. The reviewer suggested that the unrealistic large SST jump events, which could exist in C–30days experiment due to poor sampling, may play a role in forcing, instead of responding to, the atmosphere.

In response to the poor sampling issue, we examined the SST variation in 10 individual MJO events in the C–30days experiment. Figure RC#2.1 presents the SST variation of each event and composite (i.e., revised Fig. 7j) over the MC. The SST in each single event tended to be in phase but in larger amplitude in certain cases than the composite, i.e. about 3 times of SST variation (±2 K) compared to the composite that was already larger than high-frequency feedback experiments. The larger amplitudes seemed to reflect the nature of purposely designed experiment with extremely low SST feedback frequency. While the individual SST variation in signal event exhibited significant variability, the SST variation continued to adhere to the lead-lag relationship between MJO circulation and SST, highlighting a positive anomaly in the first half cycle and a subsequent negative anomaly. This result was based on the variation occurring in a limited region (i.e., 110–130° E, 5–15° S). By contrast, the SST spatial distribution could be rather unorganized. Two cases (17th and 27th year with relatively small SST amplitude) are shown in Figure RC#2.2. The unorganized structures can be identified in each event. We found the similar unorganized structure in all 10 events. The comparison suggested that the unorganized structures were not the composited results of cases with different large-scale characteristics. Reviewer's comments on SST as a forcing, instead of response, is interesting. Whereas the SST fluctuations in a MJO are largely the responses to the atmospheric forcing, its feedback (even spontaneous) in a way can be seen as a forcing. The mutual interaction (or forcing) resulted in the unique characteristics of the MJO. However, whether a larger amplitude (not seem to be jumps in Fig. RC#2.1) would induce unorghanized small-scale pertrubation is debatable. As seen in many hypothetic (or theoretical) studies, a sudden initiation of SST (or step-function like) could still induce large

scale response. We have no clear answers for the appearance of unorganiozed spatial distribution that would need purposedly designed experiments to untangle. We choose to leave the issue open in the manuscript as an unanswered question but provide the following discussion in the revised manuscript.

The reason causing the sudden change between C–24days and C–30days is not entirely clear. Two possibilities are discussed below. The first possible reason leading to this disorder is that when the ocean feedback is delayed for as long as 30 days (more than half of the MJO period), both positive and negative fluxes would contribute to the heat accumulation (or loss) in the ocean because of the MJO phase transition and result in unorganized small scale structures in ocean temperature, which could in turn affect the heat flux and convection. The second could be that the SST change become more abrupt and disrupt the large-scale nature of the MJO. However, whether large-amplitude SST fluctuations would induce unorganized small-scale pertrubation is debatable. As seen in many hypothetic (or theoretical) studies, a sudden initiation of SST (or step-function like) could induce large scale responses. This issue remains an open question that warrants further studies with purposedly designed expeiriment to untangle.

[Figure]

Figure RC#2.1 SST phase's variations of 10 single MJO events in C–30days experiment between phase 1 and 8 within the domain of 110–130° E and 5–15° S, the (a)–(j) denote MJO events from Fig. RC#2.1.

[Figure]

**Figure RC#2.2 The spatial distribution of daily-averaged SST differences is examined in two specific single MJO events by subtracting the starting date (Jan. 19) from those 30 days later, (a) SST different between Jan. 19 and Feb. 18 in 17th year (Fig. RC#2.1f); (b) SST different between Dec. 20, 27th year and Jan. 18 in 28th year (Fig. RC#2.1j).**

*RC2. My second comment is on the length of the manuscript. I feel that some figures and analyses aim to show modest sensitivities and don't explain their physical cause. I think the manuscript would benefit from being more concise and to the point.*

**Response:**

**Thank you for suggestion. We have condensed the description of experimental sensitivity, primarily addressing the physical reasons in the Conclusion and Fig. 13 and make the revised manuscript be more concise.**

---

## Author Response (AR2)

Dear GMD's Editors and Anonymous Referees:

We greatly appreciate the reviewers insightful and helpful comments regarding our manuscript. The manuscript has been revised based on reviewer's comments Below are the point-by-point replies to reviewer's comments and concerns.

Sincerely,
Yung-Yao Lan, Huang-Hsiung Hsu and Wan-Ling Tseng

**Anonymous Referee #1**
The reviewer comments are formatted in italics and the authors response to the comments are formatted in bold.
Notation *RC#1-P.* represents Reviewer Comment and Paragraph Number.

**General major/minor comments:**

> *RC#1-1. This revised manuscript has been significantly improved. Also my previous questions have been adequately addressed. I am happy to recommend this paper for publication after a very minor comment below:*
> *One of the main results from this study is: "Our results suggest that spontaneous atmosphere-ocean interaction with high vertical resolution in the ocean model is the key to the realistic simulation of the MJO and should be properly implemented in climate models." I am not quite sure what exactly is the preferred coupling frequency with the "spontaneous atmosphere-ocean interaction"? Does this refer to the 30min coupling? I would suggest to put a more explicit statement on the preferred coupling frequency for MJO modeling rather than using "spontaneous". A similar statement was also found in the conclusion part.*

**Response:**

**Thank you for the suggestion. We changed it to "more spontaneous atmosphere-ocean interaction (e.g., ocean response once every time step to every three days in this study) with high vertical resolution in the ocean model …". Please see the abstract in the revised manuscript.**

**Major Comments:**

*RC#3-1. The experiment modulates the frequency of SST change shown by the atmosphere model. In this case, SST can be abruptly changed in low-frequency SST experiments. In the C-30 days experiment, SST is changed every 30 days, and the atmosphere affects it every 30 minutes. I think the unorganized convections shown in this simulation are mostly dominated by shock from abrupt SST changes. Please revise the overall results for low-frequency SST experiments with the impact of shock from the SST change.*

**Response:**

**Reviewer's suggestion "the unorganized convections shown in this simulation are mostly dominated by shock from abrupt SST changes" is well taken and included in the revised manuscript.  Because our study did not conduct specific experiments to fully explore this effect, we mentioned in the revision that this potential effect warrants further investigation. The following discussion is added in the Conclusions section.**

**"The second possible reason would be that the SST variation in an MJO event become more abrupt and may disrupt the large-scale nature of the MJO into disorganized spatial distribution in atmosphere, ocean, and the interface where rigorous heat exchange occurs. This disrupting effect of abrupt SST variation, which is not explored in this study, warrants further studies with purposedly designed expeiment to untangle."**

**Minor Comments:**

*RC#3-2. Line 234-235: 'stationary nature of simulated MJO' has not been mentioned before. Adding (Figure 2 h-j).*

**Response:**

**We modified the sentence to "stationary nature of simulated MJO seen in Fig. 2i–j".**

*RC#3-3. Figure 9: The color of A-CTL is not recognized, especially when I print it out. I recommend changing the color for A-CTL.*

**Response:**

**Thank you for your suggestion. We changed the color for A−CTL in Fig. 9 and Fig. 11. Please see the revised manuscript for the change.**